# Cortical state transitions and stimulus response evolve along stiff and sloppy parameter dimensions, respectively

**Adrian Ponce-Alvarez[1]\*, Gabriela Mochol[1], Ainhoa Hermoso-Mendizabal[2], Jaime de la Rocha[2], Gustavo Deco[1,3,4,5]**

[1]Center for Brain and Cognition, Computational Neuroscience Group, Department of Information and Communication Technologies, Universitat Pompeu Fabra, Barcelona, Spain; [2]Institut d'Investigacions Biomèdiques August Pi i Sunyer (IDIBAPS), Barcelona, Spain; [3]Institució Catalana de la Recerca i Estudis Avançats (ICREA), Barcelona, Spain; [4]Department of Neuropsychology, Max Planck Institute for Human Cognitive and Brain Sciences, Leipzig, Germany; [5]School of Psychological Sciences, Monash University, Melbourne, Australia

**Abstract** Previous research showed that spontaneous neuronal activity presents sloppiness: the collective behavior is strongly determined by a small number of parameter combinations, defined as 'stiff' dimensions, while it is insensitive to many others ('sloppy' dimensions). Here, we analyzed neural population activity from the auditory cortex of anesthetized rats while the brain spontaneously transited through different synchronized and desynchronized states and intermittently received sensory inputs. We showed that cortical state transitions were determined by changes in stiff parameters associated with the activity of a core of neurons with low responses to stimuli and high centrality within the observed network. In contrast, stimulus-evoked responses evolved along sloppy dimensions associated with the activity of neurons with low centrality and displaying large ongoing and stimulus-evoked fluctuations without affecting the integrity of the network. Our results shed light on the interplay among stability, flexibility, and responsiveness of neuronal collective dynamics during intrinsic and induced activity.

**\*For correspondence:**
adrian.ponce@upf.edu

**Competing interests:** The authors declare that no competing interests exist.

## Introduction

How biological systems achieve a tradeoff between stability and flexibility is a central question in biology. A candidate explanation for the coexistence of these two features is *sloppiness* (*Machta et al., 2013*; *Transtrum et al., 2015*). In general, sloppiness is a property of complex models exhibiting large parameter uncertainty when fit to data, meaning that different combinations of parameters lead to a similar system behavior, while changes in some few critical parameters, called stiff parameters, significantly modifies it. In this way, biological systems can be either robust to large fluctuations of input/environmental signals which effects are embedded in a high-dimensional subspace of insensitive parameters, or, on the contrary, by tuning some few parameters, configured to be highly sensitive and selective to relevant signals.

Recently, it has been shown that the spontaneous activity of neural circuits presents sloppiness both in vitro and in vivo (*Panas et al., 2015*), suggesting that collective activity is stabilized by a subset of highly active and stable neurons, while the activity and co-activity of the remaining neurons present larger spontaneous fluctuations without strongly affecting the collective statistics. However, this view is challenged by extensive research showing that the spontaneous cortical activity transits through different synchronized and desynchronized cortical states (*Marguet and Harris, 2011*; *Harris and Thiele, 2011*; *Luczak et al., 2013*; *Pachitariu et al., 2015*) that represent statistically

different collective behaviors (*Hahn et al., 2017*) with different information processing capabilities (*Pachitariu et al., 2015*; *Engel et al., 2016*; *Beaman et al., 2017*). Moreover, how sensory inputs affect sloppiness is unknown and it is a relevant question to understand how sensory stimuli change the network state in a way that responsiveness and stability are ensured. In the present study, we examined how changes in neural network parameters correlate with spontaneous transitions among cortical states and stimulus-evoked responses.

To answer these questions, we recorded the neuronal spiking activity in the primary auditory cortex (A1) of six anesthetized rats. We analyzed the joint activity of groups of neurons while the cortex spontaneously transited through different synchronized and desynchronized cortical states and

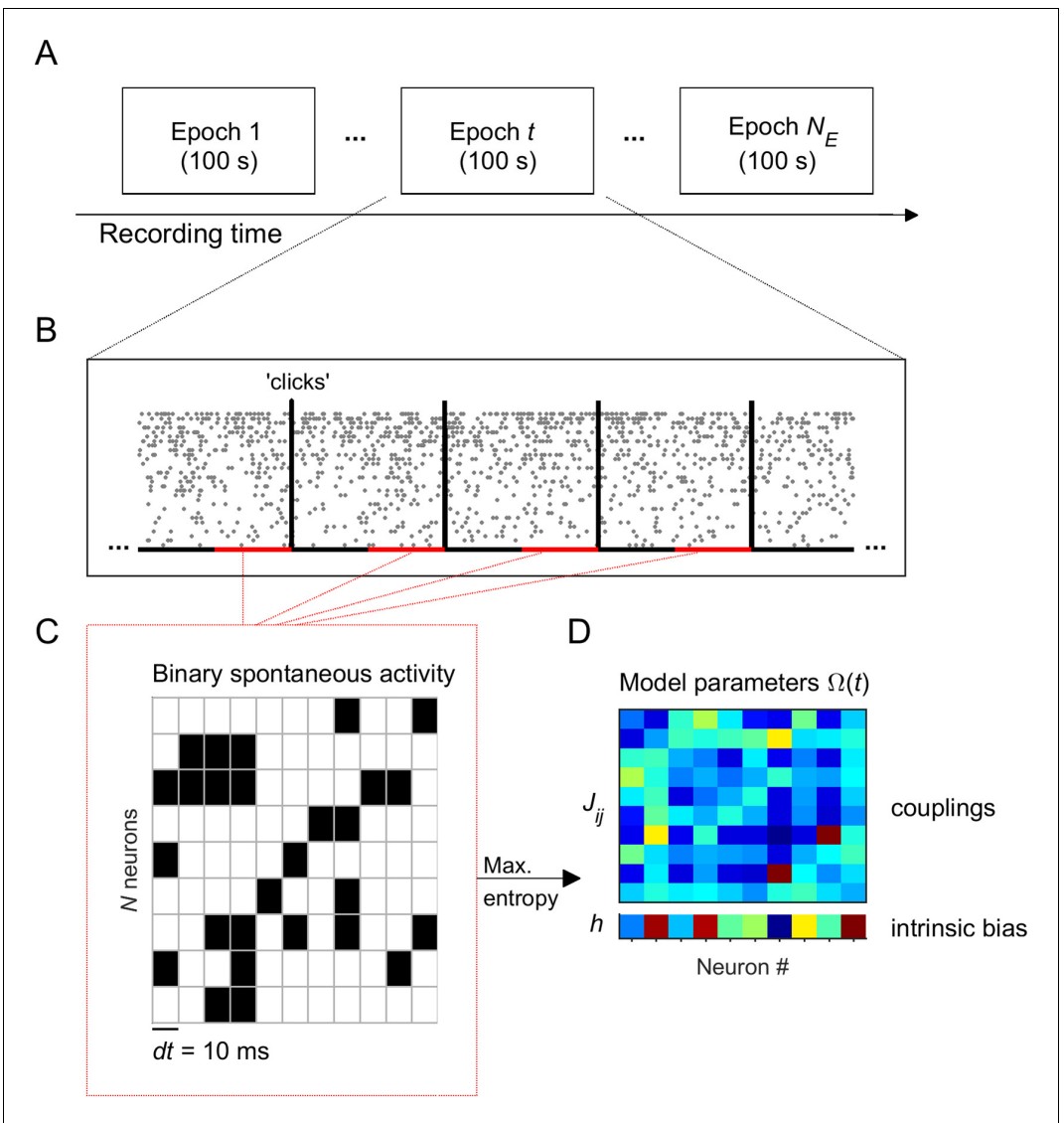

**Figure 1.** Experiment and analysis designs. (A) Each recording session was divided into $N_E$ adjacent epochs of 100 s. (B) Each epoch contained a series of stimulus presentations. Stimuli consisted on acoustic clicks. For each 100-s epoch we collected the spontaneous activity, that is the activity during 1.5-s intervals preceding each stimulus (red intervals), to build concatenated binary data. (C) Binary data was obtained by discretizing time in bins of $dt = 10$ ms. Within each time bin, the ensemble activity of $N$ neurons was described by a binary vector, $\vec{\sigma} = [\sigma_1, \sigma_2, \ldots, \sigma_N]$, where $\sigma_i = +1$ if the $i$-th neuron generated a spike (black) and $\sigma_i = -1$ otherwise (white). (D) Maximum entropy models were used to describe the binary patterns of subsets of 10 neurons, during each 100-s epoch. The model parameters $\Omega = \{h, J\}$ represents the intrinsic tendency of neuron $i$ towards activation ($\sigma_i = +1$) or silence ($\sigma_i = -1$), noted $h_i$, and the effective interaction between neurons $i$ and $j$, noted $J_{ij}$.

intermittently received external acoustic stimuli. We used a statistical model to describe the joint spiking activity with a small number of parameters. We found that the estimated parameters of neuronal ensemble activity presented sloppiness and that sensory inputs and cortical state transitions evolved in different pathways in parameter space. Specifically, we found that cortical state transitions evolve along stiff dimensions, whereas sensory-evoked activity evolves along sloppy dimensions. Finally, we showed that stiff parameters are related to the activity and co-activity of neurons with high centrality within the functional network of the recorded neurons.

## Results

We recorded spontaneous and stimulus-evoked population activity from the primary auditory cortex (A1) of urethane-anesthetized rats (n = 6) using multisite silicon microelectrodes (see Materials and methods). The data was composed of activity from $N_{pop}$ well-isolated single units ($N_{pop}$ = 44-147 neurons) and some spike-trains from multi-unit activity (3-103 spike-trains). Unless otherwise specified, the analyses present here focused on single-unit activity only. We analyzed the data during spontaneous activity and in response to acoustic 'clicks' (5-ms square pulses; inter-stimulus interval, 2.5 or 3.5 s). To track the evolution of the neuronal activity, we divided each recording session into $N_E$ adjacent epochs of 100 s, each one containing 12–29 stimulus presentations. Within each 100-s epoch the data was separated into spontaneous activity, that is the activity during 1.5-s intervals preceding each stimulus (i.e., 18–43.5 s of spontaneous activity in total for each epoch), and stimulus-evoked activity, that is the activity right after the stimulus onset (*Figure 1A–B*).

### Description of spontaneous activity patterns using maximum entropy models

We first examined the temporal evolution of the spontaneous activity across the $N_E$ epochs. Because we were interested in the evolution of the statistics of ensemble activity, we described the collective activity of groups of $N$ single-units using a maximum entropy model (MEM) (*Schneidman et al., 2006*; *Shlens et al., 2009*; *Tkačik et al., 2015*) in each epoch (see Materials and methods and *Figure 1C–D*). These models allowed us to describe the patterned activity with a small number of parameters. To fit the model, time was discretized in bins of $dt$ = 10 ms. Within each time bin, the ensemble activity of $N$ neurons was described by a binary vector, $\vec{\sigma} = [\sigma_1, \sigma_2, \ldots, \sigma_N]$, where $\sigma_i = +1$ if the $i$-th neuron fired a spike in that time bin and $\sigma_i = -1$ otherwise. The collective activity was determined by the probability distribution $P(\vec{\sigma})$ over all $2^N$ possible binary patterns. The MEM fits $P_{data}(\vec{\sigma})$ by finding a distribution $P_{MEM}(\vec{\sigma})$ that maximizes its entropy under the constraint that

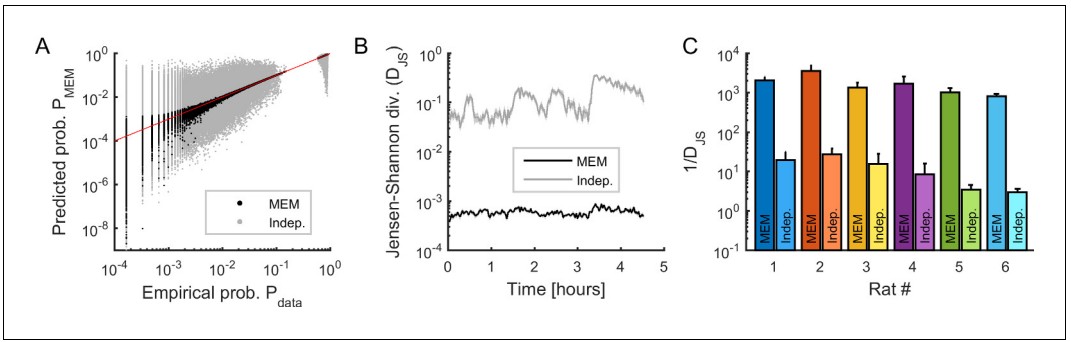

**Figure 2.** Fitting maximum entropy models (MEMs) to spontaneous activity patterns. (**A**) Comparison between the probability distribution of empirical binary patterns and the probability distribution predicted by MEMs (black dots) and independent models (gray dots), for all epochs and all neuronal ensembles. Every point represents a binary pattern that has appeared in the data at least once. Red line represents the identity line. (**B**) Jensen-Shannon divergence ($D_{JS}$) between spiking data and MEMs, and between spiking data and independent models, across time, averaged across neuronal ensembles. Error bars are smaller than the widths of the traces. Data in (**A**) and (**B**) correspond to one example rat (#1). (**C**) Goodness-of-fit ($1/D_{JS}$) for MEMs and for independent models, averaged over all models (i.e., all ensembles and all epochs), for each rat. Error bars indicate SEM.

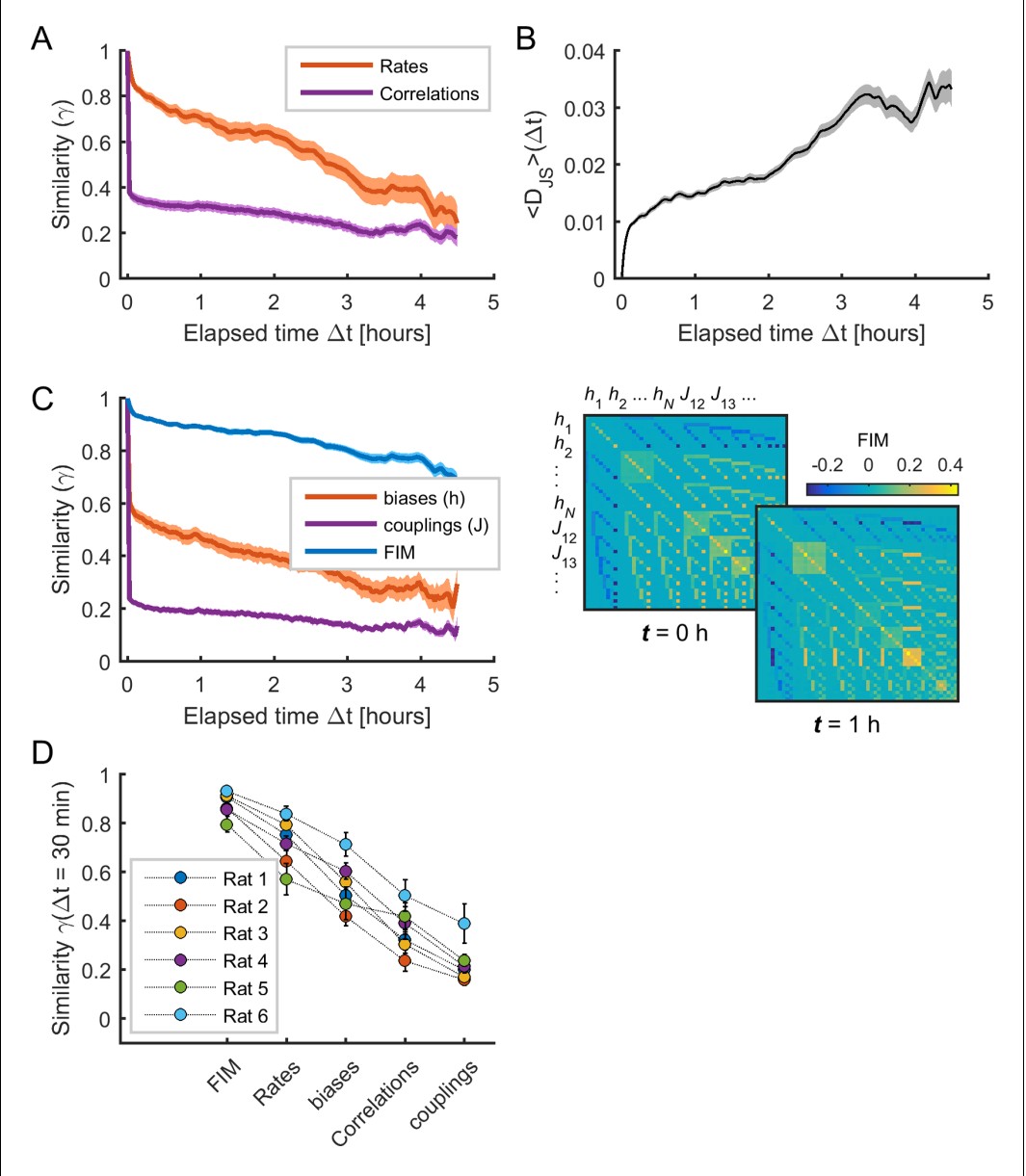

**Figure 3.** The sensitivity of model parameters was more stable than activity observables. (**A**) Similarity (i.e., Pearson correlation coefficient) of mean firing rates (red) and pairwise correlations (purple) as a function of elapsed time $\Delta t$. (**B**) Jensen-Shannon divergence ($D_{JS}$) between the distribution of empirical spiking patterns in epoch $t$ and the distribution of binary patterns of the pairwise MEM in epoch $t + \Delta t$, averaged over all $t$. (**C**) *Left*: Similarity of Fisher information matrix (FIM) elements, biases ($h_i$), and couplings ($J_{ij}$) as a function of elapsed time $\Delta t$. *Right*: FIMs at time $t = 0$ h and $t = 1$ h. Data in (**A**), (**B**), and (**C**) correspond to one example rat (#1); traces show averages over neuronal ensembles and shaded areas correspond to SEM. (**D**) Similarity of FIM elements, rates, biases, correlations, and couplings after 1/2 hour (i.e., $\Delta t = 30$ min), averaged across all neuronal ensembles, for each rat. Error bars indicate SEM.

the activation rates ($\langle \sigma_i \rangle$) and the pairwise correlations ($<\sigma_i\sigma_j>$) found in the data are preserved in the model. It is known that the maximum entropy distribution that is consistent with these constraints is the Boltzmann distribution, $P\left(\vec{\sigma}\right) \propto e^{-E\left(\vec{\sigma}\right)}$, where $E\left(\vec{\sigma}\right)$ is the energy of the pattern $\vec{\sigma}$, given by: $E\left(\vec{\sigma}\right) = -\sum_{i=1}^{N}\left[h_i\sigma_i + \frac{1}{2}\sum_{j=1}^{N} J_{ij}\sigma_i\sigma_j\right]$ (*Schneidman et al., 2006*; *Tkačik et al., 2015*). The

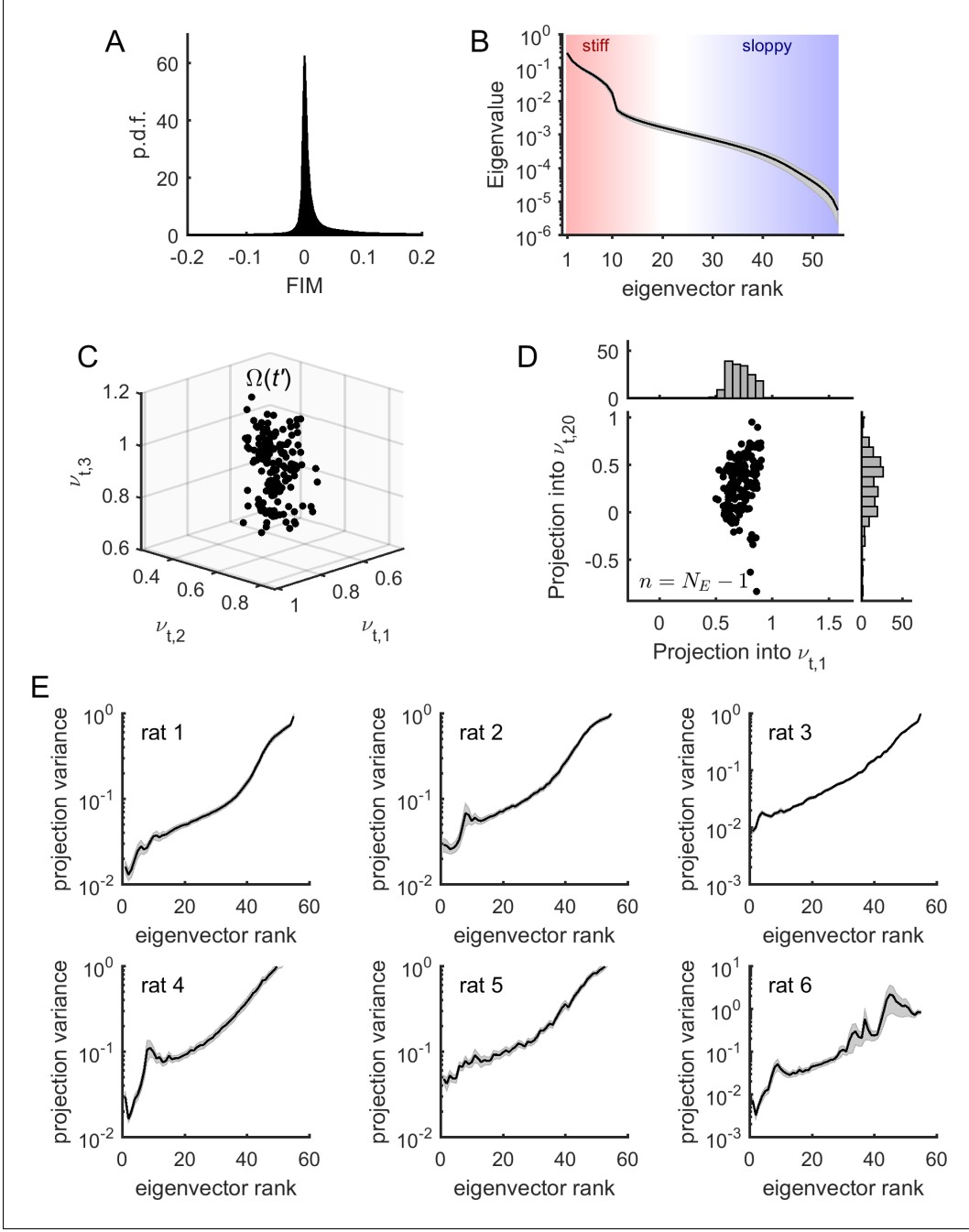

**Figure 4.** Model parameters presented sloppiness and predominantly evolved along sloppy dimensions. (A) Distribution of FIM elements, for all epochs, all neuronal ensembles, and all rats. (B) Eigenvalues of the FIM, average across epochs and neuronal ensembles, for an example rat (# 1). Shaded areas represent SEM. Stiff and sloppy dimensions correspond to FIM eigenvectors of lowest and highest ranks, respectively. (C) Projection of $\Omega(t')$ into the first three eigenvectors of the FIM from a given epoch $t$, for all $t' \neq t$. Data from one neuronal ensemble from rat 1. (D) Projection of $\Omega(t')$ into the first and the 20th eigenvectors of the FIM from epoch $t$, noted $\nu_{t,1}$ and $\nu_{t,20}$, respectively, for all $t' \neq t$. *Top inset*: distribution of projections into $\nu_{t,1}$. *Right inset*: distribution of projections into $\nu_{t,20}$. Note higher variance of projections into $\nu_{t,20}$ than into $\nu_{t,1}$. Data from one neuronal ensemble from rat 1. (E) Average variance of projections of $\Omega(t')$ into the different eigenvectors of the FIM at epoch $t$ (for all $t' \neq t$), for the different rats. Traces represent average over neuronal ensembles and shaded areas represent SEM. The online version of this article includes the following figure supplement(s) for figure 4:

**Figure supplement 1.** Parameters projections into FIM eigenvectors: data *vs.* stationary surrogates.

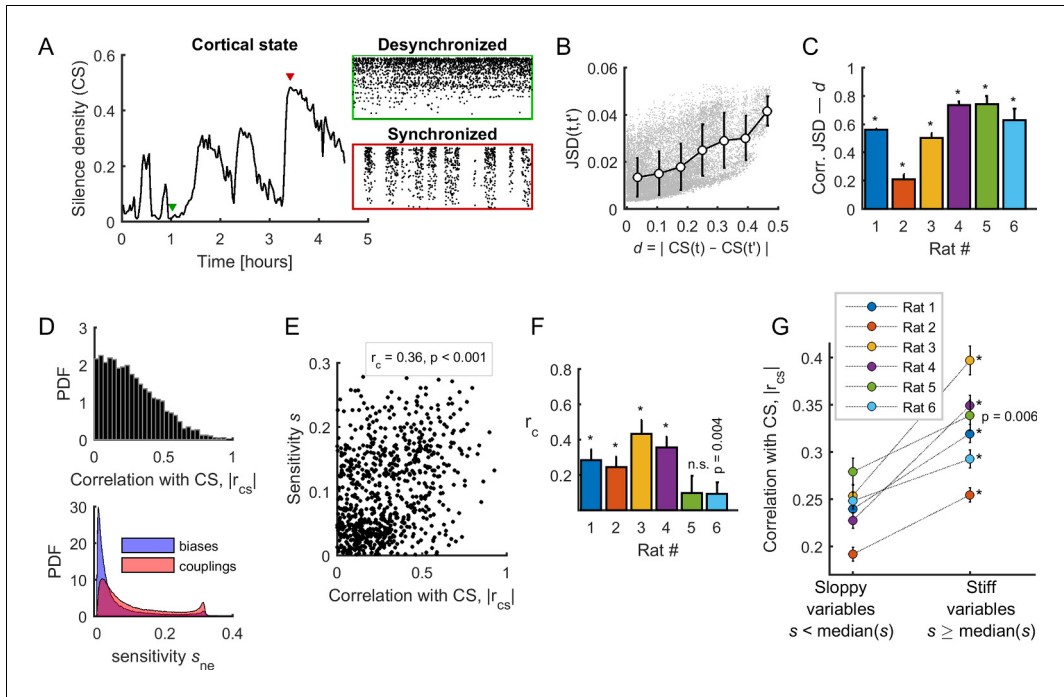

**Figure 5.** Spontaneous neuronal activity evolved along stiff dimensions during cortical state transitions. (**A**) Silence density was used to characterize the cortical state. *Green inset*: low values of the silence density indicate desynchronized cortical states (each row represents the spike train of a single-unit). *Red inset*: high values of the silence density indicate synchronized cortical states. Data from rat 1. (**B**) Difference in collective pattern statistics, that is $D_{JS}$, between different epochs, $t$ and $t'$, as a function of the corresponding difference in silence density, noted $d$. Each gray dot corresponds to a pair of epochs $(t, t')$. The solid line indicates the average relation between $D_{JS}$ and $d$; error bars indicate SD. Data from all neuronal ensembles from rat 1. (**C**) Correlation coefficient between $D_{JS}$ and $d$, for all rats. *: p < 0.001. (**D**) *Top*: Distribution of the absolute value of the correlations between the cortical state and the activity observables, noted $|r_{cs}|$. *Bottom*: Distribution of parameter sensitivity values for biases ($h$ parameters) and couplings ($J$ parameters), for all models from all rats. (**E**) $|r_{cs}|$ vs. sensitivity $s$ of all activity variables (i.e., firing rates and pairwise correlations). Correlation: $r_c = 0.36$; p <0.001. Data from rat 4 ($N_{pop} = 72$). (**F**) Correlation between $|r_{cs}|$ and the sensitivity for each dataset. *: p < 0.001. Error bars indicate correlation 95% confidence interval. (**G**) $|r_{cs}|$ for sloppy and stiff variables. *: p < 0.001, paired $t$-test. Error bars indicate SEM. The online version of this article includes the following figure supplement(s) for figure 5:

**Figure supplement 1.** Alternative definition of sensitivity.

model parameter $h_i$ represents the intrinsic tendency of neuron $i$ towards activation ($\sigma_i = +1$) or silence ($\sigma_i = -1$) and the parameter $J_{ij}$ represents the effective interaction between neurons $i$ and $j$. Once we learned the parameters $\Omega = \{\boldsymbol{h}, \boldsymbol{J}\}$ using a gradient descent algorithm (see Materials and methods), the expected probability of any pattern is known. For each recording session and for each of the $N_E$ epochs, we fitted the model using the spontaneous binarized activity from an ensemble of $N = 10$ randomly selected single neurons from the entire population of $N_{pop}$ single neurons. We chose $N = 10$ because 100-s epochs provided around 5000 observed spontaneous patterns, which is a reasonable amount to get an estimate of the distribution of the $2^{10} = 1024$ possible patterns. To accurately estimate models of larger $N$, the epochs ought to be much larger preventing possibility to investigate the temporal evolution of the model along the experiment. We finally repeated the process of randomly choosing $N = 10$ single units $Q$ times for each experiment (for datasets 3 and 5: $Q = 10$ ensembles, otherwise: $Q = 20$). In summary, for each recording session, we built $Q \times N_E$ models, each composed of 10 units. Before studying the evolution of the model parameters $\Omega(t)$ across epochs ($t = 0, 1, 2..$), we first evaluated how well the MEM fitted the data.

For each epoch, we used the Jensen-Shannon divergence ($D_{JS}$, see Materials and methods) to measure the similarity between the probability distribution of the empirical and model binary patterns (*Figure 2A–C*). We compared this similarity to the distribution of binary patterns predicted

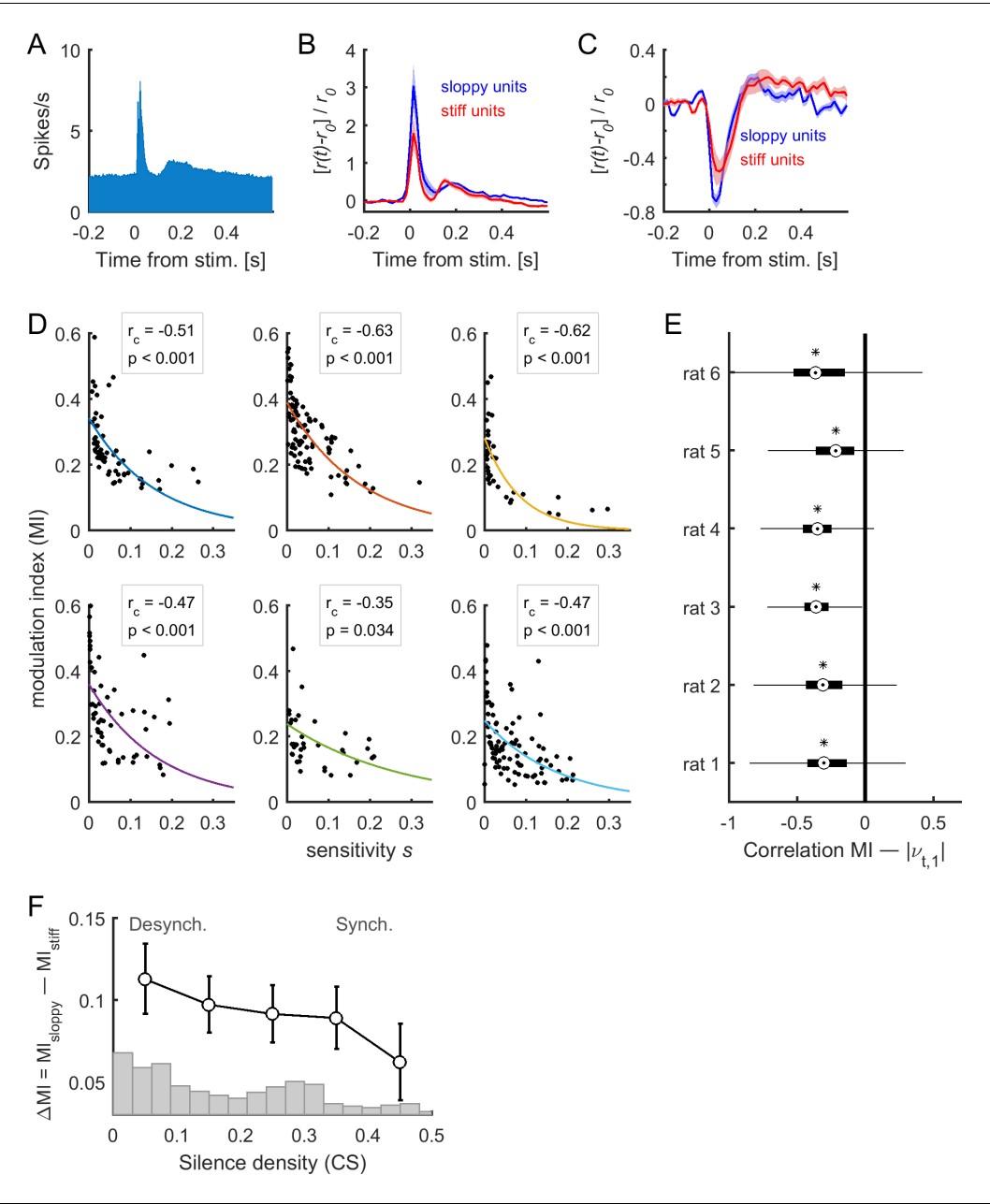

**Figure 6.** Stimulus-evoked neuronal activity evolved along sloppy dimensions. (**A**) Population responses to acoustic clicks. (**B–C**) Median-split of sensitivity *s* was used to separate stiff neurons and sloppy neurons. The mean responses for stiff and sloppy neurons are shown in the case of excited responses (**B**) and suppressed responses (**C**). The responses were normalized by the average pre-stimulus activity $r_0$. Shaded areas correspond to SEM. Data in (**A**), (**B**), and (**C**) correspond to one example rat (#1). (**D**) Modulation index (MI) as a function of associated sensitivity of firing rates ($s_i$, with $1 \leq i \leq N_{pop}$), for each dataset. Each dot corresponds to a single neuron of the recorded population. The correlation between MI and sensitivity was negative for all datasets ($r_c$: correlation coefficient; p: p-value). Solid lines indicate exponential fits. (**E**) Correlation between MI(*t*), calculated in epoch *t*, and $|\nu_{t,1}|$, for all neuronal ensembles of each of the rats. On each box, the central mark indicates the median, and the bottom and top edges of the box indicate the 25th and 75th percentiles, respectively. Asterisks indicate significantly negative medians (p < 0.001, two-sided signed rank test). (**F**) Difference of the MI of sloppy neurons minus the MI of stiff neurons as a function of cortical state (i.e., silence density), averaged for all rats (black trace; error bars indicate SEM). The gray bars indicate the distribution of silence density values.

The online version of this article includes the following figure supplement(s) for figure 6:

**Figure supplement 1.** Alternative definition of sensitivity.

from independent-MEMs, for which only the activation rates were preserved (i.e., only $h$ was optimized). We found that the empirical distribution was well approximated by MEMs and that, for all recording sessions, the goodness-of-fit (i.e., $1/D_{JS}$) was orders of magnitude higher for MEMs than for independent-MEMs (*Figure 2C*), leading to excellent model performances (i.e., Kullback-Leibler ratio equal to $0.95 \pm 0.03$ on average, see Materials and methods).

## Temporal evolution of activity observables, model parameters, and their sensitivity

We next analyzed the temporal evolution of the different spiking data statistics and the model parameters. We first measured the temporal variation of the activity observables (i.e., firing rates and pairwise correlations) by calculating the average Pearson correlation (or similarity $\gamma$; see Materials and methods) between the values in epoch $t$ and those in epoch $t + \Delta t$ (*Figure 3A*). This similarity rapidly decayed with $t$, indicating that the observables substantially changed over time. We next examined how much these variations influenced the evolution of the collective activity characterized by the distribution of binary patterns. For this, we evaluated how well the data in a given epoch $t$ could be explained by the MEM constructed using the data at time $t + \Delta t$. Specifically, we calculated $\langle D_{JS} \rangle (\Delta t)$, given by the average Jensen-Shannon divergence between the distribution of data binary patterns in epoch $t$, that is $P_{\text{data},t}$, and the distribution of binary patterns predicted by the MEM constructed using the data in epoch $t + \Delta t$, that is $P_{\text{MEM},t+\Delta t}$ (see Materials and methods). We found that $\langle D_{JS} \rangle (\Delta t)$ increased as a function of $t$, indicating that the collective activity changed during the recording session, so that the model parameters $\Omega(t)$ at epoch $t$ did not predict the collective activity at epoch $t + \Delta t$ (*Figure 3B*). Indeed, the model parameters substantially changed over time, with a rapidly decaying similarity (*Figure 3C*, *orange* and *purple* traces).

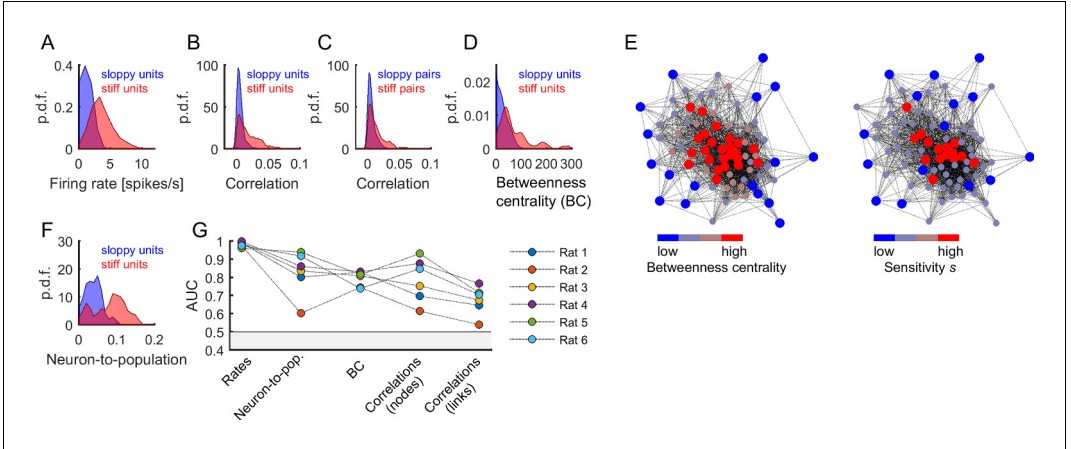

**Figure 7.** Central and highly active neurons were associated to stiff parameters. (A) Distribution of firing rates of sloppy and stiff neurons. (B) Distribution of correlations among sloppy neurons and among stiff neurons. (C) The distribution of correlations was also calculated for the links (i.e, pairs of neurons with associated parameters $J_{ij}$). Note that, in principle, links can be related to pairs composed of one sloppy and one stiff neuron. (D) Distribution of betweenness centrality of sloppy and stiff neurons. (E) Connectivity graph: each node represents a neuron and links represent significant correlations between pairs of neurons. The graph was plotted using force-directed layout, that is using attractive forces between strongly connected nodes and repulsive forces between weakly connected nodes. *Left*: the nodes were colored as a function of betweenness centrality. *Right*: the nodes were colored as a function of associated sensitivity $s$. Note the high overlap between both color labeling methods, indicating that sensitivity was highly predictive of the centrality of the nodes. (F) Distribution of neuron-to-population couplings of sloppy and stiff units. Panels A–F show data from rat 1. (G) Area under the receiver operating curve (AUC) quantifying the separation of distributions of sloppy and stiff classes. All AUC values were significantly higher than 0.5 (p < 0.001).

The online version of this article includes the following figure supplement(s) for figure 7:

**Figure supplement 1.** Alternative definition of sensitivity.

As shown in *Panas et al. (2015)*, changes in model parameters can differently contribute to collective activity, since the model can be sensitive to changes in some few combinations of parameters. Following this, we next evaluated the sensitivity of model parameters by calculating the Fisher information matrix (FIM, see Materials and methods) for each neuronal ensemble and each epoch. The FIM measures how much the model log-likelihood $P_{\text{MEM}}\left(\vec{\sigma}|\Omega\right)$ changes with respect to changes in the parameters $\Omega$. We first notice that the FIM had the highest stability across time, compared to the data firing rates and correlations and the model parameters (*Figure 3C*, *blue* trace). Indeed, the similarity after 1/2 hour was $\gamma = 0.882 \pm 0.002$ for the FIM, $\gamma = 0.732 \pm 0.003$ for the firing rates, $\gamma = 0.551 \pm 0.004$ for the biases, $\gamma = 0.364 \pm 0.004$ for the correlations, $\gamma = 0.234 \pm 0.003$ for the couplings ($F_{4,495}$ = 305.73, p<0.001, one-way ANOVA followed by Tukey's post hoc analysis) (*Figure 3D*). Altogether these results show that the sensitivity of the model parameters remained relatively stable despite substantial changes in firing rates, correlations, collective activity and the model parameters themselves.

## Spontaneous neuronal activity presents sloppiness

Having shown that the sensitivity of model parameters was relatively stable during the recording sessions, we next studied the structure of the FIMs. First, we noted that most elements of the FIM had near-zero values (*Figure 4A*) indicating that most of the parameters had a small effect on the model log-likelihood. In contrast, a small fraction of elements had values strongly different from zero as revealed by the heavy tail of the distribution of FIM values (*Figure 4A*). To identify the parameter combinations that had the strongest effect on model behavior, we decomposed the FIM into eigenvectors and classified them according to their eigenvalue (*Figure 4B*). We observed that, except for some few eigenvalues, most of the FIM eigenvalues were small, corresponding to combinations of parameters that had little effect on model behavior. These unimportant parameter combinations defined the sloppy dimensions of the model. The few eigenvectors with large eigenvalues defined the stiff parameter dimensions along which the model behavior was strongly affected.

In the following we showed that the temporal evolution of the model parameters occurred predominantly along the sloppy dimensions. For this, we projected the parameters $\Omega(t')$, calculated at time $t'$, into the eigenvectors of the FIM at time $t$, denoted $\nu_{t,1}$, $\nu_{t,2}$, ..., $\nu_{t,k}$, ..., where $k$ is the rank of the eigenvector (*Figure 4C*). For each dimension, or eigenvector, we obtained a distribution of projections of parameters $\Omega(t')$ (*Figure 4D*). To quantify how much the parameters varied along each eigenvector, we calculated the average variance of each projection as a function of the rank of the eigenvector. We found that the projection variance increased as a function of the eigenvector's rank for all datasets (*Figure 4E*). This indicates that the model parameters predominantly evolved along sloppy dimensions (i.e., FIM eigenvectors of highest rank $k$), while they remained relatively stable along stiff dimensions (i.e., FIM eigenvectors of lowest rank $k$). Using stationary surrogate data, we controlled that these parameter fluctuations were not fully explained by estimation errors and, furthermore, that parameter fluctuations along sloppy dimensions were those that deviated the most from the stationary case (see *Figure 4—figure supplement 1*). Nevertheless, we noted that the projection variance into the stiff dimensions, albeit small, was not zero. This means that the model also evolved along parameter dimensions that had a strong impact on the collective activity. We hypothesized that changes in collective behavior, associated to changes in stiff parameters, were related to changes in cortical state.

## Cortical state transitions evolve along stiff dimensions

To test this hypothesis, we first measured the cortical state in each epoch $t$ using silence density, $CS(t)$, defined as the fraction of 20-ms time bins with zero population activity, that is no spikes from any neuron (see Materials and methods) (*Luczak et al., 2013*; *Pachitariu et al., 2015*; *Mochol et al., 2015*). To obtain the most accurate estimate of silence density, we used all the spikes from the merge of all the single-units and multi-units in the calculation of $CS(t)$. During the course of the experiment, we observed large fluctuations in silence density, with low and high values associated to desynchronized and synchronized cortical states, respectively (*Figure 5A*). We found that differences in collective dynamics in different epochs, quantified by $D_{JS}(t,t') = D_{JS}\left(P_{\text{data},t}; P_{\text{data},t'}\right)$, significantly co-varied with the changes in cortical state, given by $d = |CS(t) - CS(t')|$ (averaged correlation

coefficient 0.56 ± 0.08, p < 0.001) (*Figure 5B–C*). Thus, changes in collective behavior correlated with changes in cortical state.

We next asked which activity observables, that is the firing rate of each neuron and all pairwise correlations, related more to cortical state transitions. For this, we calculated the absolute correlation, $|r_{cs}|$, between the cortical state $CS(t)$ and the activity observables. We found that $|r_{cs}|$ was broadly distributed between 0 and 0.94, thus some observables correlated more with the cortical state (*Figure 5D*, top panel). Next, to relate the sensitivity of model parameters (their stiffness) to the activity observables, we measured the sensitivity of a given parameter by its average contribution to the first eigenvector of the FIM and we associated it to the corresponding observable (*Panas et al., 2015*). We defined the sensitivity $s_{ne}$ at the neuronal ensemble level and the sensitivity $s$ at the population level (see Materials and methods). Note that the ranges of the sensitivity of biases (*h*) and couplings (*J*) were similar (*Figure 5D*, bottom panel), and that sensitivities calculated in the first and the second halves of the recording session were highly correlated (correlation coefficient > 0.82, for all rats; average: 0.89 ± 0.03). We found a significant positive correlation between the associated sensitivity (*s*) and the correlation with the cortical state ($|r_{cs}|$) in 5/6 datasets (*Figure 5E–F*). Thus, the observables that correlated more with the cortical state were those with the highest associated sensitivity. This result led us to separate the activity observables into two classes, called 'sloppy' and 'stiff', based on whether the associated sensitivity (*s*) was lower or higher than the median of *s*. We found that stiff variables were significantly more correlated with the cortical state than the sloppy variables (p<0.01 for all datasets, paired *t*-test; *Figure 5G*). This relationship was preserved when using an alternative, more general definition of sensitivity that considered the contribution to all eigenvectors of the FIM, instead of the contribution to the first eigenvector only (see *Figure 5—figure supplement 1*). Altogether, these results indicate that neuronal activity and co-activity preferentially evolved along sensitive (stiff) parameter dimensions during cortical state transitions.

## Sensory-evoked activity evolves along sloppy dimensions

The above results indicate that, although intrinsic spontaneous dynamics predominantly evolved along sloppy dimensions (*Figure 4F*), cortical state transitions were governed by changes in stiff parameters (*Figure 5G*). We next investigated which parameter dimensions were explored when the neural network was driven by external sensory inputs, that is during stimulus-evoked activity (*Figure 6A*). We observed that evoked responses (which could be increased or decreased with respect to pre-stimulus baseline firing rate) were larger for sloppy neurons than for stiff neurons (*Figure 6B–C*). To quantify the responsiveness of each neuron, we calculated the modulation index (MI, see Materials and methods) of each neuron in response to acoustic stimuli. We next calculated the relation between MI, calculated during evoked activity, and the sensitivity *s* associated to firing rates, calculated during the spontaneous activity as above. We found that the more responsive neurons were those with the lowest associated sensitivity (*Figure 6D–E*). This indicates that stimulus-evoked neuronal activity evolved mostly along sloppy dimensions. This result was replicated when using a more general definition of sensitivity that considered the contribution to all eigenvectors of the FIM (see *Figure 6—figure supplement 1A*). Finally, we evaluated the difference, noted ΔMI, between the MI of sloppy and stiff neurons as a function of cortical state $CS(t)$. Specifically, first, the MI values in each epoch were averaged according to different ranges of the silence density. Second, the MI values of sloppy and stiff neurons were compared within each range. We found that ΔMI was maximal during desynchronized activity, and minimal during synchronized activity (*Figure 6F*, see also *Figure 6—figure supplement 1B*). Thus, the cortical activity during stimulus response evolved predominantly along sloppy dimensions for the desynchronized cortical state, while, in the synchronized state, the dominance of sloppy fluctuations was reduced, and stiff fluctuations became comparable.

Finally, correlations between cortical state fluctuations and sensitivity and between MI and sensitivity could reflect a dependency between sensitivity and model estimation errors. To test this, we evaluated the mean error on the model estimation of the observables and test its interaction with sensitivity, cortical state fluctuations, and MI (see Appendix 1 and *Appendix 1—figure 1*). We found that model estimation errors correlated with sensitivity, but they could not fully explain neither the positive correlation between sensitivity and cortical state nor the negative correlation between sensitivity and MI.

## Stiff parameters were associated to central neurons within the neuronal network

In this section, we further investigate the properties of neurons and pairs of neurons with respect to their associated parameter sensitivity. As above, we separated the neurons and pairs of neurons into two classes, called 'sloppy units/pairs' and 'stiff units/pairs', based on whether the associated sensitivity ($s$) was lower or higher than the median $s$ (units were associated to parameters $h_i$, and pairs or links were associated to parameters $J_{ij}$). With this dichotomization, we found that stiff units were significantly more active than sloppy units (*Figure 7A*). We quantified this by performing receiver operating characteristic (ROC) analysis and used the area under the ROC curve (AUC) as a measure of how well the firing rates distributions of the two classes were separated (AUC = 0.961–0.998, p < 0.001, for all rats; *Figure 7G*). Stiff neurons were also significantly more correlated among them than sloppy neurons (AUC = 0.615–0.932, p < 0.001, for all rats; *Figure 7B,G*). The distributions of correlations remained well separated when calculated for the links, that is pairs of neurons with associated parameters $J_{ij}$ (AUC = 0.541–0.766, p < 0.001, for all rats; *Figure 7C,G*).

To further investigate the structure of correlations, we evaluated the centrality of stiff and sloppy neurons within the observed network of neurons. For this we used the betweenness centrality (BC), a measure of node centrality in a graph or network, which in our case was given by the functional connectivity matrix among the recorded neurons (see Materials and methods). The BC measures the extent to which a node in the graph tends to lay on the shortest path between other nodes. Thus, a node with higher BC has more influence over the network, because more information passes through that node. We found that stiff neurons had significantly more centrality in the functional connectivity graph than sloppy neurons (AUC = 0.740–0.831, p<0.001, for all rats; *Figure 7D,G*). This indicates that stiff neurons were part of the core of the graph, while sloppy neurons were part of the graph periphery, as clearly shown using graph visualization (*Figure 7E*; *Fruchterman and Reingold, 1991*). BC values were correlated with firing rates (correlation coefficient: 0.59 ± 0.11), which could suggest that differences in BC between stiff and sloppy neurons were simply a consequence of differences in firing rates. However, using surrogate data that preserved the observed firing rates and produced correlations through global modulations, we found that neither the structure of correlations nor the BC values could be trivially predicted by globally modulated firing rates but they were rather suggestive of functional interactions (see Appendix 1 and *Appendix 1—figure 2*). Thus, in addition to different firing rates, different correlations and BC values were supplementary features of stiff and sloppy neurons.

Moreover, previous work has shown that cortical neurons differ in their coupling to the population activity, with neurons that activate most often when many others are active and neurons that tend to activate more frequently when others are silent (*Okun et al., 2015*). Thus, along with centrality, we calculated the neuron-to-population coupling, given by the Pearson correlation between the activity of each neuron $i$ and the number of coactive neurons (excluding neuron $i$; see Materials and methods). We found that stiff neurons were significantly more coupled to the population activity than sloppy neurons (AUC = 0.603–0.939, p < 0.001, for all rats; *Figure 7F,G*). In summary, stiff units were more active, more central, more coupled among them, and more coupled to the population activity than sloppy units. The same results were found when using a more general definition of sensitivity that considered the contribution to all eigenvectors of the FIM (*Figure 7—figure supplement 1*).

## Discussion

We here studied the changes in activity caused by intrinsic (i.e. cortical state) and extrinsic (i.e., stimulus-evoked) sources in A1 neuronal ensembles in an estimated parameter space. The parameter space was obtained using the maximum entropy principle, providing a handful number of parameters describing the probability of all possible binary activity patterns. These parameters differed in their impact on collective activity that was sensitive to a few combinations of parameters, called stiff dimensions, but insensitive to many others called sloppy dimensions. Our results suggest that spontaneous cortical state transitions and stimulus-driven activity evolved along different parameter dimensions. Indeed, in one hand, while most of the fluctuations during spontaneous activity evolved along sloppy dimensions, some residual ongoing fluctuations evolved along stiff dimensions, and these fluctuations were correlated with synchronized/desynchronized cortical state transitions. On

the other hand, stimulus-induced activity was larger in sloppy dimensions than in stiff dimensions, an effect that was most prominent during the desynchronized cortical state. Note that the observation that both spontaneous and stimulus-driven activities predominantly evolve along sloppy dimensions results from the strong similarity of spontaneous and evoked activity, reported in several previous studies (*Arieli et al., 1996*; *Kenet et al., 2003*; *MacLean et al., 2005*; *Luczak et al., 2009*). Finally, by classifying the neurons as stiff versus sloppy neurons (i.e., those contributing more or less to the principal stiff dimension) we found that the firing rates and the functional connectivity topology significantly differed between the two classes of neurons. It should be noted, however, that, since sensitivity is a continuous variable, the two classes of neurons that we defined here do not represent two disjoint groups but rather represent two parts of a continuum.

The observation that stimulus-induced activity evolved along sloppy dimensions can have important functional implications. It suggests that a stimulus can modulate the activity of a subset of sloppy neurons without entirely affecting the collective activity. This could be an efficient functional architecture to encode sensory information without perturbing other ongoing or memory-stored processes. Consistent with this view and with previous studies (*Margolis et al., 2012*; *Mizuseki and Buzsáki, 2013*; *Panas et al., 2015*), our results suggest that the integrity of the network is ensured by a core of highly active stiff neurons, which have strong functional connections among them (either through anatomical connections or common inputs), while topologically peripheral sloppy neurons (within the functional connectivity graph) can be largely modulated by external inputs. A similar subnetwork of highly active, interconnected neurons has been recently identified in the mice neocortex (*Yassin et al., 2010*). Importantly, sensory input was not required to drive these cells. Previous studies of complex systems have derived general principles of core/periphery network structures: the network periphery is more variable, evolvable, and plastic than the network core, while the network core facilitates system robustness (*Kitano, 2004*; *Csermely et al., 2013*). Thus, we hypothesize that sloppy neurons could also be more affected by synaptic plasticity, allowing for network reconfiguration without loss of stability. Consistent with this, previous work on whole-brain fMRI has observed core stability and peripheral flexibility over the course of learning (*Bassett et al., 2013*), and recent analyses of functional networks from calcium imaging data recorded in mouse primary auditory cortex revealed a stable core and a variable periphery over time (*Betzel et al., 2019*). Furthermore, we observed that stimulus responses evolved more pronouncedly along sloppy dimensions in the desynchronized state, while in the synchronized state fluctuations along sloppy and stiff dimensions were comparable (*Figure 6F*). This supports the view that responses along sloppy dimensions provide information processing benefits, since previous studies have shown that auditory stimuli in rodents (*Marguet and Harris, 2011*; *Pachitariu et al., 2015*) and visual stimuli in both rats (*Goard and Dan, 2009*) and monkeys (*Beaman et al., 2017*) are better represented in the desynchronized state as compared to the synchronized state.

The properties of spontaneous and induced cortical dynamics observed in the present anesthetized condition are likely to be relevant also during wakefulness. Indeed, several studies reported the existence of synchronized cortical states during wakefulness (for review see *Zagha and McCormick, 2014*), and global fluctuation resembling transitions between up and down periods during alert or quiescent wakefulness (*Petersen et al., 2003*; *Luczak et al., 2007*; *Poulet and Petersen, 2008*; *Zagha et al., 2013*; *Tan et al., 2014*; *Engel et al., 2016*) or even during task engagement (*Sachidhanandam et al., 2013*). Moreover, sloppiness has been observed in asynchronous spontaneous activity under light anesthesia (*Panas et al., 2015*), we thus expect to observe a similar stiff-sloppy architecture in the awake state. However, we believe that the comparison of Fisher information matrices during wakefulness and during different levels of anesthesia could provide valuable information about the principles governing vigilance.

We found that stiff neurons were more linked to the observed neuronal population activity than sloppy neurons. Stiff neurons had higher centrality in the functional connectivity graph and higher coupling to the population activity than sloppy neurons. Previous research showed that neurons differ in their coupling to the population activity, with neurons that activate most often when many others are active, called 'choristers', and neurons that tend to activate more frequently when others are silent, called 'soloists' (*Okun et al., 2015*). Our results suggest that stiff and sloppy neurons are chorister and soloist neurons, respectively. In other words, changes in the activity of stiff/chorister neurons lead to changes in collective behavior (i.e., cortical states), while the activity of sloppy/soloist neurons can spontaneously fluctuate or respond to stimuli without strongly affecting the collective

behavior. Thus, we believe that the roles of stiff/chorister neurons and sloppy/soloist neurons are important to understand tradeoffs between responsiveness and stability of the network. Furthermore, we here studied the evolution of neuronal activity on the time scale of hours and found that fluctuations on stiff parameter dimensions were the weakest and were related to cortical state transitions, which time scale is in the order of tens of minutes (*Hahn et al., 2017*; *Mochol et al., 2015*). Previous studies have reported prominent changes on neuronal activity and tuning properties over days, but with stable decoding performances of population activity (*Chestek et al., 2007*; *Ziv et al., 2013*; *Panas et al., 2015*). However, we hypothesize that learning or adaptation to changing environments could lead to large changes in collective activity. In that case, particular attention could be paid to the influence of high-order areas on the activity of subsets of stiff and sloppy neurons from sensory areas, as top-down regulation might be a mechanism to control the stabilizing network core.

The existence of cortical neurons with different sensitivities (from sloppy to stiff neurons) provides new valuable architectural constrains for models of the brain state and its transitions. Several past studies have modeled the synchronized brain dynamics as transitions between two attractors. Depending on the model specificity those transitions could be noise driven (*Mejias et al., 2010*; *Mochol et al., 2015*; *Jercog et al., 2017*) or caused by some fatigue mechanism (*Compte et al., 2003*; *Hill and Tononi, 2005*; *Mattia and Sanchez-Vives, 2012*). To make the system works in a desynchronized regime it was enough to increase the background input to the network (*Bazhenov et al., 2002*; *Hill and Tononi, 2005*; *Curto et al., 2009*; *Destexhe, 2009*; *Mochol et al., 2015*). Given our present results, the models could be extended to include a network core/periphery architecture, a non-homogeneous background input preferentially targeting the network core, and different stimulus spatial distributions. Such a model would provide insights on the interplay between cortical state transitions and sensory representation. Moreover, our findings question the view that the mechanisms by which background and stimulus inputs impact the dynamics are similar, as assumed in the simple bi-stable rate model (*Mochol et al., 2015*).

Finally, we here described the patterned activity of small ($N = 10$) neuronal ensembles using MEMs. It is known that MEMs of small sizes can present departures from the observed distribution of summed activities and higher-order correlations (*Tkačik et al., 2014*). Recent advancements on learning algorithms allow to construct MEMs of ~100 neurons. However, these models cannot be used in a time-resolved manner, as we did here, due to limited data in each epoch. Small model sizes are thus the cost to pay to study the evolution of collective activity over time in a meaningful time scale (i.e., the one of cortical state transitions).

# Materials and methods

## Key resources table

| Reagent type (species) or resource | Designation | Source or reference | Identifiers | Additional information |
|---|---|---|---|---|
| Biological sample (*Sprague–Dawley rat*) | Sprague–Dawley rat | https://doi.org/10.1073/pnas.1410509112 | | six rats, 250–400 g |
| Software, algorithm | Matlab | MathWorks | RRID:SCR_001622 | All analyses |
| Software, algorithm | Klustakwik | http://klustakwik.sourceforge.net/ | RRID:SCR_014480 | Spike sorting (detection and initial clustering) |
| Software, algorithm | EToS | http://etos.sourceforge.net/ | | Spike sorting (detection and initial clustering) |
| Software, algorithm | Klusters | http://neurosuite.sourceforge.net/ | | Spike sorting (clustering) |

## Ethics statement

All experiments were carried out in accordance with protocols approved by the Animal Ethics Committee of the University of Barcelona (Comité d'Experimentació Animal, Universitat de Barcelona, Ref 116/13).

## Experimental techniques

We analyzed the neuronal activity recorded in the primary auditory cortex (A1) of 6 anesthetized rats (Sprague–Dawley; 250–400 g). The experimental procedures and spikes sorting procedures have been previously described in *Mochol et al. (2015)*. Briefly, rats were anesthetized with urethane (1.5

g/kg body weight) and silicon microelectrodes (Neuronexus) with 32 or 64 channels were inserted in deep layers (depth, 600–1,200 µm) of the primary auditory cortex. The spiking activity from single units and multi-units (i.e., neurons that were not well isolated) was simultaneously recorded during spontaneous activity and in response to acoustic 'clicks' (5 ms square pulses; interstimulus interval, 2.5 or 3.5 s; see *Table 1*). In some datasets, double clicks (5 ms square pulses; 50- or 100 ms inter-click interval) were also presented, but, in the present study, we analyzed only the responses to single click. The spiking data is publicly available here: https://github.com/adrianponce/Spont_stim_spiking_A1.

## Cortical state

Long continuous recordings (mean, ~2 h) were divided into $N_E$ 100-s epochs, and cortical state was estimated in each epoch based on spontaneous pooled population activity, that is the merge of single and multiunit spike trains during the 1.5-s intervals preceding each stimulus presentation. Cortical state was quantified using silence density defined as the fraction of 20-ms time bins with no population activity. Silent and active periods were obtained from the merge of consecutive empty and nonempty bins, respectively.

## Maximum entropy models

The spontaneous spiking activity of ensembles of $N$ single neurons was studied using statistical modeling based on maximum entropy principle. The ensemble activity was binarized in non-overlapping time bins of $dt$ = 10 ms, during which neuron $i$ either did ($\sigma_i = +1$) or did not ($\sigma_i = -1$) generate one or more spikes. The state of the neural ensemble is described by a binary pattern $\vec{\sigma} = [\sigma_1, \sigma_2, \ldots, \sigma_N]$, and thus the collective activity is described by the probability distribution $P(\vec{\sigma})$ over all $2^N$ possible binary patterns. We estimated $P(\vec{\sigma})$ using a Maximum entropy model (MEM). The MEM finds $P(\vec{\sigma})$ by maximizing its entropy under the constraint that some empirical statistics are preserved. A pairwise-MEM provides a solution under the constraint that the activation rates ($<\sigma_i>$) and the pairwise correlations ($<\sigma_i\sigma_j>$) are preserved. The maximum entropy distribution $P(\vec{\sigma})$ that is consistent with these expectation values is given by the Boltzmann distribution (*Schneidman et al., 2006*; *Tkačik et al., 2015*):

$$P(\vec{\sigma}) = \frac{e^{-E(\vec{\sigma})}}{\sum_{\{\vec{\sigma}\}} e^{-E(\vec{\sigma})}}, \tag{1}$$

where $E(\vec{\sigma})$ is the energy of the pattern $\vec{\sigma}$, given by:

$$E(\vec{\sigma}) = -\sum_{i=1}^{N} h_i\sigma_i - \frac{1}{2}\sum_{i=1}^{N}\sum_{j=1}^{N} J_{ij}\sigma_i\sigma_j, \tag{2}$$

and $Z = \sum_{\{\vec{\sigma}\}} e^{-E(\vec{\sigma})}$ is the partition function.

**Table 1.** Number of neurons (SU: single-units, MU: multi-unit), number of 100 s epochs, number of stimulus presentations in 100 s epochs, and number of neuronal ensembles, for each dataset.

| | No. of neurons | No. of 100 s epochs | Stimulus presentations in 100 s epochs | No. of neuronal ensembles (Q) |
|---|---|---|---|---|
| Rat 1 | SU: 81; MU: 3 | 163 | 12–14 | 20 |
| Rat 2 | SU: 147; MU: 13 | 74 | 12–14 | 20 |
| Rat 3 | SU: 44; MU: 30 | 70 | 12–20 | 10 |
| Rat 4 | SU: 72; MU: 103 | 59 | 10–20 | 20 |
| Rat 5 | SU: 58; MU: 39 | 29 | 28–29 | 10 |
| Rat 6 | SU: 112; MU: 83 | 28 | 17–29 | 20 |

The model parameter $h_i$, called intrinsic bias, represents the intrinsic tendency of neuron $i$ towards activation ($\sigma_i = +1$) or silence ($\sigma_i = -1$) and the parameter $J_{ij}$ represents the effective interaction between neurons $i$ and $j$. The estimation of the model parameters $\Omega = \{h, J\}$ was achieved through a gradient descent algorithm (see below). For each recording session, we constructed models for $Q$ ensembles of $N = 10$ randomly selected single neurons from the entire population of $N_{\mathrm{pop}}$ single neurons and learned the model parameters using the spontaneous binarized activity within each 100-s epoch. Thus, for each recording session, we built $Q \times N_E$ models of 10 units. We were interested on the evolution of the model parameters over time, that is $\Omega(t)$. Note that, for a given model, the number of free parameters is the sum of intrinsic biases and effective couplings, $N + N(N - 1)/2 = 55$, that is $\Omega = [h_1, h_2, \ldots, h_N, J_{12}, J_{13}, \ldots]$.

## Estimation of MEM parameters

The MEM parameters $\Omega = \{h, J\}$ were iteratively adjusted to minimize the absolute difference between the empirical activation rates ($\langle \sigma_i \rangle$) and correlations ($\langle \sigma_i \sigma_j \rangle$) and those ($\langle \sigma_i \rangle_{\mathrm{model}}$, $\langle \sigma_i \sigma_j \rangle_{\mathrm{model}}$) predicted by the model through Metropolis Monte Carlo simulations (100,000 samples). Specifically, each iteration is given by: $h_i^{new} = h_i^{old} - \alpha(\langle \sigma_i \rangle_{\mathrm{model}} - \langle \sigma_i \rangle)$, and $J_{ij}^{new} = J_{ij}^{old} - \alpha\left(\langle \sigma_i \sigma_j \rangle_{\mathrm{model}} - \langle \sigma_i \sigma_j \rangle\right)$, where $\alpha$ is the learning rate ($\alpha = 0.1$). In our study we stopped the re-estimations once the differences between the empirical and model values are less than a tolerance threshold (0.005) or if this tolerance was not reached within a maximum number of iterations (100).

## MEM goodness-of-fit

The goodness-of-fit of the MEMs was evaluated using the Jensen–Shannon divergence ($D_{JS}$) between the probability distribution of the empirical and model binary patterns (**Marre et al., 2009**). $D_{JS}$ is a symmetric version of the Kullback-Leibler divergence ($D_{KL}$) and is given as:

$$D_{JS}(P_{\mathrm{data}}; P_{\mathrm{MEM}}) = \frac{1}{2}D_{KL}\left[P_{\mathrm{data}}; \frac{(P_{\mathrm{data}} + P_{\mathrm{MEM}})}{2}\right] + \frac{1}{2}D_{KL}\left[P_{\mathrm{MEM}}; \frac{(P_{\mathrm{data}} + P_{\mathrm{MEM}})}{2}\right],$$  (3)

Where $P_{\mathrm{MEM}}$ was given by the Boltzmann distribution of the model, $P_{\mathrm{data}}$ was estimated from the $N$-dimensional binary patterns observed in the data, and:

$$D_{KL}(P_1; P_2) = \sum_{\{x\}} P_1(x) \log \frac{P_1(x)}{P_2(x)}.$$  (4)

The fitting of MEM (second-order model) was compared to the fit obtained using independent-MEM, that is in which only for which only the activation rates ($<\sigma_i>$) are preserved (i.e., only $h$ is optimized; first-order model). In this case, the pattern energy is given by: $E\left(\vec{\sigma}\right) = -\sum_{i=1}^{N} h_i \sigma_i$.

Furthermore, the performance of the model can be evaluated using the Kullback-Leibler ratio, $R$ (**Shlens et al., 2009**). This ratio is given by comparing the Kullback-Leibler divergence between the distribution $P_1$ of the first-order model (i.e., independent-MEM) and the distribution of the actual data, $D_1 = D_{KL}(P_1; P_{\mathrm{data}})$, with the Kullback-Leibler divergence between the distribution $P_2$ of the second-order model and the distribution of the actual data, $D_2 = D_{KL}(P_2; P_{\mathrm{data}})$. Specifically, the Kullback-Leibler ratio is defined as:

$$R = \frac{D_1 - D_2}{D_1}.$$  (5)

This ratio can range between 0 and 1, with one giving the highest performance.

## Fisher information matrix

Because in the MEM, all the information about the collective activity is contained in the probability distribution of the binary patterns, $P\left(\vec{\sigma}\right)$, one can define the model parameter space as $P\left(\vec{\sigma}|\Omega\right)$. We were interested in knowing which parameters, or combination of parameters, have a strong effect on the collective activity. To measure how distinguishable two models, with parameters $\Omega$ and

$\Omega + \delta\Omega$, are based on their predictions, we used the Fisher information matrix (FIM). Indeed, the Kullback-Leibler divergence between the two models can be written as:

$$D_{KL}(\Omega; \Omega + \delta\Omega) = FIM_{kl}\delta\Omega_k\delta\Omega_l + \mathcal{O}(\delta\Omega^3), \tag{6}$$

where $1 \leq k, l \leq 55$, and the matrix $FIM$ is given by:

$$FIM_{kl} = \sum_{\{\vec{\sigma}\}} P(\vec{\sigma}|\Omega) \frac{\partial \log P(\vec{\sigma}|\Omega)}{\partial \Omega_k} \frac{\partial \log P(\vec{\sigma}|\Omega)}{\partial \Omega_l}. \tag{7}$$

The FIM represents the curvature of the log-likelihood of the model, $\log P(\vec{\sigma}|\Omega)$, with respect to the model parameters. It quantifies the sensitivity of the model to changes in parameters. By calculating the eigenvalues of the FIM, we can determine which combinations of parameters affect the most the model's behavior.

In the case of MEM, the FIM can be easily obtained by using *Equations 1, 2, and 7*. As a result, the FIM is given by the covariance matrix of observables associated to the parameters which can be calculated from the model through Metropolis Monte Carlo simulations (500,000 steps), that is:

$$FIM_{kl} = \langle x_k x_l \rangle - \langle x_k \rangle \langle x_l \rangle, \tag{8}$$

with $1 \leq k, l \leq 55$ and $\vec{x} = [\sigma_1, \sigma_2, \ldots, \sigma_N, \sigma_1\sigma_2, \sigma_1\sigma_3, \ldots]$.

## Sensitivity measures

The FIM was calculated for every neuronal ensemble at every 100-s epoch and it was decomposed into eigenvectors, noted $v_{t,1}, v_{t,2}, \ldots, v_{t,k}, \ldots$, where $k$ is the rank of the eigenvector and $t$ denotes the epoch. Following *Panas et al. (2015)*, within each neuronal ensemble, we measured the sensitivity of a given parameter by its averaged contribution to the first eigenvector of the FIM, that is the sensitivity of $i$-th parameter is given by $s_{\text{ne},i} = \frac{1}{N_E}\sum_t |v_{t,1}(i)|$, with $1 \leq i \leq N$.

We next constructed a sensitivity measure for the entire population of $N_{\text{pop}}$ neurons. For this, we defined the set of all single neuron indices and all pairs of neurons $\boldsymbol{I} = \{1, \ldots, N_{\text{pop}}, (1,2), (1,3), \ldots\}$. This set has $L = N_{\text{pop}} + \frac{N_{\text{pop}}(N_{\text{pop}}-1)}{2}$ elements. For each element $j$ of $\boldsymbol{I}$, we defined the sensitivity $s_j$ as the average of $s_{\text{en},i}$ over the neuronal ensembles that contained the $j$-th single neuron or the pair of neurons (i.e., those neuronal ensembles for which $i$ maps to $j$). In other words, $s_{\text{ne}}$ denotes the sensitivity within an ensemble of $N = 10$ neurons and has 55 elements, and $s$ denotes the sensitivity within the entire population of $N_{\text{pop}}$ neurons and has $L$ elements. This allows comparison of $s$ with statistics derived from the population of $N_{\text{pop}}$ neurons.

Parameters that contributed less to the first eigenvector could in principle contribute to the other stiff dimensions (those with lower rank $k$, e.g., $k = 2$). For this reason, we also considered an alternative definition of sensitivity that considers the weighted contribution to all eigenvectors of the FIM. For each neuronal ensemble and each 100-s epoch $t$, we defined the weighted sensitivity of the parameter $i$ as the temporal average of its contribution to the eigenvectors of the FIM, weighted by the associated eigenvalues ($a_{t,1}, \ldots, a_{t,55}$):

$$s_{\text{ne},i}^w = \frac{1}{N_E}\sum_{t=1}^{N_E}\sum_{k=1}^{55} \frac{a_{t,k}|v_{t,k}(i)|}{a_{t,1} + a_{t,2} + \ldots + a_{t,55}}. \tag{9}$$

As previous, from $s_{\text{ne}}^w$ one can construct a weighted sensitivity $s^w$ at the population level.

Finally, we separated the activity observables into two classes, called "sloppy" and "stiff", based on whether the associated sensitivity $s$ was lower or higher than the median sensitivity.

## Similarity measures

Temporal variations of model parameters and data statistics were quantified using the average correlation between the parameters/statistics at time t and the parameters/statistics at time $t + \Delta t$. For example, let $\vec{r}(t)$ the average firing rates of the neurons during the epoch $t$, the similarity measure is given by:

$$\gamma(\Delta t) = \frac{1}{N_E - \Delta t} \sum_{t=1}^{N_E - \Delta t} \rho\left[\vec{r}(t), \vec{r}(t+\Delta t)\right], \tag{10}$$

Where $N_E$ is the number of epochs and $\rho$ is the Pearson correlation coefficient. In the case of FIM, the matrix was vectorized to calculate $\rho$.

To evaluated how well the data in a given epoch $t$ could be explained by the MEM constructed using the data at time $t + \Delta t$. Specifically, we defined the similarity measure $\langle D_{JS}\rangle(\Delta t)$, given by the average Jensen-Shannon divergence between the distribution of data binary patterns in epoch $t$, that is $P_{\text{data},t}$, and the distribution of binary patterns predicted by the MEM constructed using the data in epoch $t + \Delta t$, that is $P_{\text{MEM},t+\Delta t}$. This measure is given as:

$$\langle D_{JS}\rangle(\Delta t) = \frac{1}{N_E - \Delta t} \sum_{t=1}^{N_E - \Delta t} D_{JS}\left(P_{\text{data},t}; P_{\text{MEM},t+\Delta t}\right). \tag{11}$$

In other words, $1/\langle D_{JS}\rangle(\Delta t)$ quantifies how well, on average, the model with parameters $\Omega(t + \Delta t)$ represents the data from epoch $t$.

## Modulation index

We quantified the responsiveness of the neurons to sensory stimuli through the modulation index (MI) defined as:

$$MI = \frac{\left|r_{stim} - r_{spon}\right|}{r_{stim} + r_{spon}}, \tag{12}$$

where $r_{spon}$ is the pre-stimulus average spike count, calculated in the 0.5-s pre-stimulus interval, and $r_{stim}$ is the average spike count calculated from stimulus onset to 0.5 s after stimulus onset. With this definition, strongly increased or suppressed stimulus responses, with respect to pre-stimulus activity, lead to high MI values.

## Betweenness centrality

For each recording session, we analyzed the network defined by the Pearson correlation matrix of the activities of all single units. The centrality of a neuron, or node, within the network was quantified using the betweenness centrality (BC) measure. BC is given by the number of shortest paths that pass through a given node. The correlation matrix was compute for all 100-s epochs and, for each matrix element, we tested whether the mean of the $N_E$ correlation values differs from 0 ($t$ test followed by Bonferroni correction), resulting in a binary graph $G$ with entries equal to 1 if correlation were significantly different from zero (corrected p-value < 0.05) and 0 otherwise. The BC for each node of the graph was given by:

$$BC(i) = \sum_{k \neq i \neq l} \frac{p(kl; i)}{p(kl)} \tag{13}$$

where $p(kl)$ is the total number of shortest paths from node $k$ to node $l$ and $p(kl; i)$ is the number of those paths that pass through $i$.

## Neuron-to-population coupling

To quantify the coupling of each neuron to the activity of the neuronal population, we calculated, for each epoch, the Pearson correlation between the activity of each neuron ($\sigma_i$) and the number of coactive neurons (i.e., with $\sigma_i = +1$) at each time bin ($dt$ = 10 ms) from the neuronal population of single units (without including the neuron $i$). The neuron-to-population coupling was given by the average of the correlation coefficient across epochs.

## ROC analysis

We used the receiver operating characteristic curve (ROC) to evaluate the separation between the distributions of observables from sloppy and stiff classes. Let $X_{\text{sloppy}}$ and $X_{\text{stiff}}$ be the sloppy variables, that is those variables with associated sensitivity ($s$) lower than the median $s$, and the stiff variables,

that is those variables with associated sensitivity ($s$) higher than the median $s$, respectively. The ROC curve, $f(c)$, is build by plotting the probability of $P(X_{\text{sloppy}}>c)$ against the probability of $P(X_{\text{stiff}}>c)$, for each all $c$. The area under the ROC curve (AUC) is a measure of separation between $P(X_{\text{sloppy}})$ and $P(X_{\text{stiff}})$, and it is given by:

$$AUC = \int f(c)dc. \tag{14}$$

AUC ranges between 0 and 1, with AUC = 0 if $P(X_{\text{sloppy}})$ and $P(X_{\text{stiff}})$ are completely separated and $X_{\text{sloppy}}>X_{\text{stiff}}$, AUC = 1 if $P(X_{\text{sloppy}})$ and $P(X_{\text{stiff}})$ are completely separated and $X_{\text{stiff}}>X_{\text{sloppy}}$, and AUC = 0.5 if $P(X_{\text{sloppy}})$ and $P(X_{\text{stiff}})$ are undistinguishable. We used a permutation test (1000 re-samples), in which observables and classes were randomly associated, to assess AUC values that were significantly different from 0.5.

### Stationary surrogates

To construct the stationary surrogates we first randomly selected a reference epoch $t$. Second, we generated binary data using the MEM estimated from the spiking data at this reference epoch, that isusing parameters $\Omega(t)$, through Monte Carlo simulations of the model to obtain 5000 binary patterns. Third, we repeated the Monte Carlo simulations $N_E$ times. Finally, for each of the $N_E$ pieces of surrogate data, we estimated new MEM parameters, $\Omega'$, using gradient descend, and we calculated the corresponding Fisher Information Matrix (FIM) using 500,000 Monte Carlo steps as described above. By construction, the obtained surrogate data were stationary and had the same length of the original spiking data. Thus, parameter fluctuations in the surrogate data were only due to model estimation errors.

## Acknowledgements

APA and GD received funding from the FLAG-ERA JTC (PCI2018-092891). GD acknowledges funding from the European Union's Horizon 2020 FET Flagship Human Brain Project under Grant Agreement 785907 HBP SGA2, the Spanish Ministry Project PSI2016-75688-P (AEI/FEDER) and the Catalan Research Group Support 2017 SGR 1545. GM was supported by a Juan de la Cierva fellowship (IJCI-2014–21937) from the Spanish Ministry of Economy and Competitiveness. AHM received support from the Spanish Ministry of Economy and Competitiveness (BES-2011–049131). JR received funding from the Spanish Ministry of Economy and Competitiveness together with the European Regional Development Fund Grants SAF2010-15730 and SAF2013-46717-R. We thank L Hollender for sharing her data.

## Additional information

### Funding

| Funder | Grant reference number | Author |
| --- | --- | --- |
| European Commission | Flag-Era JTC PCI2018-092891 | Adrian Ponce-Alvarez<br>Gustavo Deco |
| Ministerio de Economía y Competitividad | Juan de la Cierva Fellowship IJCI-2014-21937 | Gabriela Mochol |
| European Regional Development Fund | SAF2013-46717-R | Jaime de la Rocha |
| Ministerio de Economía y Competitividad | BES-2011-049131 | Ainhoa Hermoso-Mendizabal |
| Ministerio de Economía y Competitividad | SAF2010-15730 | Jaime de la Rocha |
| European Regional Development Fund | SAF2010-15730 | Jaime de la Rocha |
| Ministerio de Economía y Competitividad | SAF2013-46717-R | Jaime de la Rocha |

| Horizon 2020 Framework Programme | 785907 HBP SGA1 | Gustavo Deco |
|---|---|---|
| Horizon 2020 Framework Programme | 785907 HBP SGA2 | Gustavo Deco |
| Ministerio de Economía y Competitividad | PSI2016-75688-P | Gustavo Deco |
| Catalan Research Group Support | 2017 SGR 1545 | Gustavo Deco |

The funders had no role in study design, data collection and interpretation, or the decision to submit the work for publication.

### Author contributions

Adrian Ponce-Alvarez, Conceptualization, Formal analysis, Investigation, Methodology; Gabriela Mochol, Data curation, Formal analysis, Investigation, Methodology; Ainhoa Hermoso-Mendizabal, Data curation, Formal analysis, Methodology; Jaime de la Rocha, Data curation, Methodology; Gustavo Deco, Funding acquisition, Methodology

### Author ORCIDs

Adrian Ponce-Alvarez (iD) https://orcid.org/0000-0003-1446-7392
Jaime de la Rocha (iD) http://orcid.org/0000-0002-3314-9384

### Ethics

Animal experimentation: All experiments were carried out in accordance with protocols approved by the Animal Ethics Committee of the University of Barcelona (Comité d'Experimentació Animal, Universitat de Barcelona, Reference: 116/13).

### Decision letter and Author response

Decision letter https://doi.org/10.7554/eLife.53268.sa1
Author response https://doi.org/10.7554/eLife.53268.sa2

## Additional files

### Supplementary files
• Transparent reporting form

### Data availability

We made the spiking data publicly available at: https://github.com/adrianponce/Spont_stim_spiking_A1.

The following dataset was generated:

| Author(s) | Year | Dataset title | Dataset URL | Database and Identifier |
|---|---|---|---|---|
| Ponce-Alvarez A, Mochol G, Hermoso-Mendizabal A, delaRocha J, Deco G | 2019 | Spont_stim_spiking_A1 | https://github.com/adrianponce/Spont_stim_spiking_A1 | github, Spont_stim_spiking_A1 |

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

# Appendix 1

## Estimation errors and sensitivity

Correlations between cortical state fluctuations and parameter sensitivity and between MI and sensitivity could reflect a dependency between sensitivity and model estimation errors. To test this, we here evaluated the error on the model estimation of the observables (firing rates and correlations) as a function of sensitivity. For each epoch $t$, the firing rates and covariances estimated by the MEM can be calculated from the model parameters by means of the Boltzmann distribution. The estimated moments $\langle\sigma_i\rangle$ and $\langle\sigma_i\sigma_j\rangle$ are given by:

$$\langle\sigma_i\rangle = \sum_{\{\vec{\sigma}\}} P\left(\vec{\sigma}\right)\sigma_i, \tag{15}$$

$$\langle\sigma_i\sigma_j\rangle = \sum_{\{\vec{\sigma}\}} P\left(\vec{\sigma}\right)\sigma_i\sigma_j. \tag{16}$$

The estimated firing rates and the covariances are given as $r_i = (\langle\sigma_i\rangle + 1)/(2dt)$ and $C_{ij} = \langle\sigma_i\sigma_j\rangle - \langle\sigma_i\rangle\langle\sigma_j\rangle$ (*Appendix 1—figure 1A-B*). We found that the firing rates and covariances estimated from the data and those estimated by the MEM highly correlated (average correlation coefficient for rates: $0.999 \pm 0.001$; for covariances: $0.956 \pm 0.018$, *Appendix 1—figure 1C*), indicating acceptable fits of the models. Indeed, the model performances, evaluated through the Kullback-Leibler ratio $R$, were also high (average equal to $0.95 \pm 0.03$, *Appendix 1—figure 1D*).

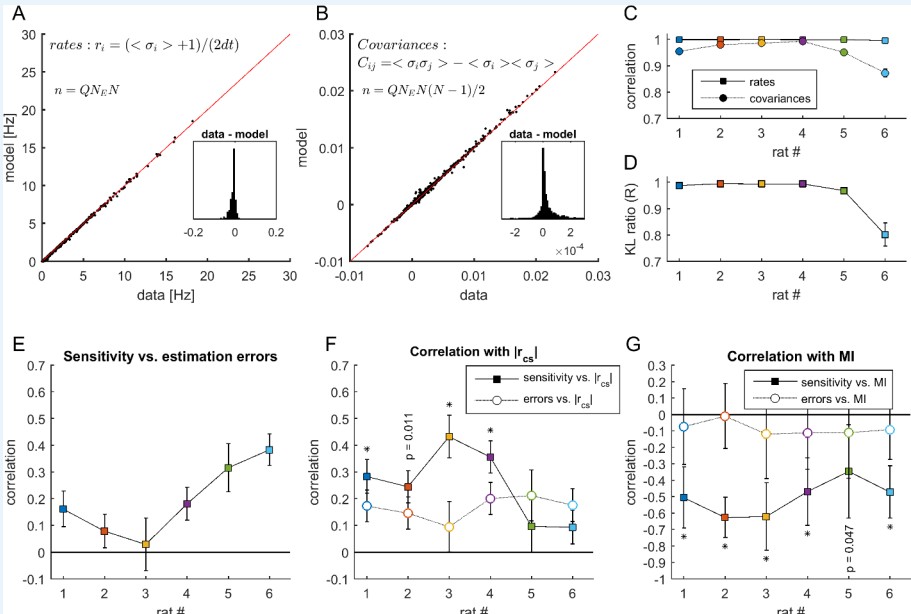

**Appendix 1—figure 1.** Estimation errors and sensitivity. (**A-B**) Comparison between the firing rates and covariances measured in the data and those estimated by the MEM. Data from rat 4. Each point represents the activity of a neuron (in panel **A**) or co-activity of a pair of neurons (in panel **B**) from a given neuronal ensemble and at a given epoch. The data of all neuronal ensembles and all epochs are presented. Insets show the resulting differences between statistics estimated from the data and from the model. (**C**) Correlation between the firing rates and covariances measured in the data and those estimated by the MEM, for each dataset. (**D**) Average Kullback-Leibler ratios for all learned models of each dataset. Error bars indicate SEM. (**E**) Correlation between sensitivity $s$ and estimation errors $Err$. (**F**) Full squares:

correlation between sensitivity and $|r_{cs}|$ (same values as in *Figure 5F*). Open circles: correlation between estimation errors and $|r_{cs}|$. For each dataset the correlation coefficients were compared, given the observed correlations in panel E using Meng's z-test for dependent correlations, *: p < 0.01. (**G**) Full squares: correlation between sensitivity of firing rates and MI (same values as in *Figure 6D*). Open circles: correlation between firing rate estimation errors and MI. *: p < 0.01, Meng's z-test for dependent correlations. In panels (**C**) and (**E**–**G**), error bars indicate correlation 95% confidence interval.

We next defined the estimation errors as the averaged squared difference between the values estimated from the data and those estimated by the model:

$$EE[r_i] = \frac{1}{N_E} \sum_{t=1}^{N_E} \frac{\Delta r_{t,i}}{\sigma_{\Delta r}}, \tag{17}$$

$$EE[C_{ij}] = \frac{1}{N_E} \sum_{t=1}^{N_E} \frac{\Delta C_{t,i,j}}{\sigma_{\Delta C}}, \tag{18}$$

where $\Delta r_{t,i} = \left( r_i^{\mathrm{data,t}} - r_i^{\mathrm{model,t}} \right)^2$ and $\Delta C_{t,i,j} = \left( C_{ij}^{\mathrm{data,t}} - C_{ij}^{\mathrm{model,t}} \right)^2$, and $\sigma_{\Delta r}$ and $\sigma_{\Delta r}$ denote the standard deviations of $\Delta r_{t,i}$ and $\Delta C_{t,i,j}$, respectively —this normalization is needed because $\Delta r_{t,i}$ and $\Delta C_{t,i,j}$ had different magnitudes. Using the same procedure as for sensitivity $s$, we obtained, for each dataset, a variable $Err$ that represented the mean estimation error of the $N_{\mathrm{pop}}$ neurons and all pairwise combinations.

Since estimation errors correlated with sensitivity (average: $0.192 \pm 0.055$; *Appendix 1— figure 1E*), we asked whether the correlations between sensitivity and cortical state observed in *Figure 5F* could be fully predicted by correlations between errors and cortical state. We found that correlations between sensitivity $s$ and $|r_{cs}|$ were higher than correlations between estimation errors and $|r_{cs}|$ ($0.252 \pm 0.056$ vs. $0.168 \pm 0.017$ on average; *Appendix 1—figure 1F*). We used Meng's z-test for dependent correlations to compare the correlations between sensitivity $s$ and $|r_{cs}|$ and those between estimation errors and $|r_{cs}|$, given the correlation between $s$ and errors, and found significant differences in 4/6 datasets (p < 0.05). This was the case for datasets with the highest model performances (datasets 1–4).

Finally, we tested whether estimation errors for firing rates correlated with the modulation index (MI) of the neurons and compared the results with those obtained in *Figure 6D*. We found that correlations between estimation errors and MI were substantially weaker than correlations between estimation errors and MI ($-0.506 \pm 0.043$ vs. $-0.086 \pm 0.017$ on average; p<0.05 for all datasets, Meng's z-test for dependent correlations; *Appendix 1—figure 1G*). We concluded that, although estimation errors correlated with sensitivity, they did not fully explain neither the correlation between sensitivity and cortical state (for the datasets with highest model performance) nor the correlation between sensitivity and MI.

## Non-homogeneous Poisson process surrogate data

In *Figure 7A–D* we found that stiff neurons were more active, more coupled among them, and more central than sloppy neurons. We here investigated the possibility that correlations and BC values could be trivially predicted by firing rates. We tested whether the structure of correlation could arise from spontaneous global fluctuations increasing or decreasing the firing of the neurons, instead of functional connectivity among them. For this, we constructed surrogate data that preserved the mean firing rate of the neurons, and the number and duration of silent (i.e., no spikes, empty 20 ms time bins) and active (i.e., non-empty 20 ms time bins) periods in each epoch, but without other structured correlations among the units.

To do this, we generated non-homogeneous Poisson process (NHPP) surrogate data as follows. We first calculated the average spontaneous firing rates of the single neurons during active periods in the original data, $\vec{r_e} = \left[ r_{e,1}, r_{e,2}, \ldots, r_{e,N_{pop}} \right]$, given by the number of spikes, in a given epoch $e$, divided by the total duration of active periods within this epoch. During silent periods the firing rate of all single units was zero. Next, for each unit $i$ and within each epoch

$e$, we generated non-homogeneous Poisson spike trains using the estimated firing rate as the intensity $\lambda_{e,i}(t)$ of the point process, that is $\lambda_{e,i}(t)$ was a step function with $\lambda_{e,i}(t) = r_{e,i}$, for times $t$ within active periods, and $\lambda_{e,i}(t) = 0$, otherwise. The resulting surrogate data preserved the firing rate of each neuron and silence periods of population activity (**Appendix 1—figure 2A, E**).

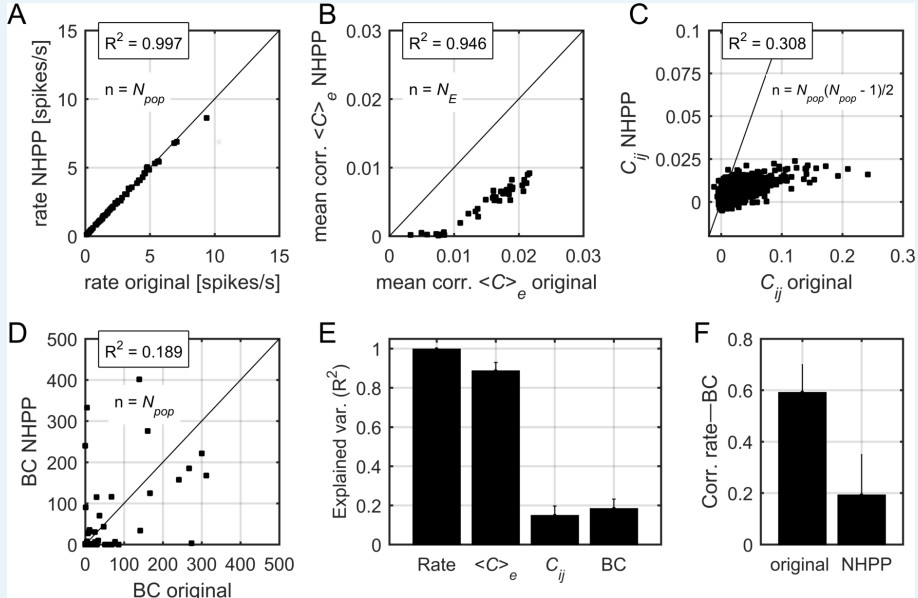

**Appendix 1—figure 2.** Correlations and centrality in non-homogeneous Poisson process surrogate data. (**A**) Relation between firing rates in the original data and firing rates in the non-homogeneous Poisson process (NHPP) surrogates. $R^2$ indicates the fraction of explained variance. The black line indicates the identity line. (**B**) Relation between the mean pairwise correlation in the original data and that in NHPP surrogates. Each point represents the mean correlation (averaged over neurons) in a given epoch. (**C**) Relation between pairwise correlations, averaged over the $N_E$ epochs, in the original data and those in NHPP surrogates. Each point represents a pair of neurons. (**D**) Relation between BC values in the original data and those in NHPP surrogates (see Materials and methods for calculation of BC). Data in (**A**), (**B**), (**C**), and (**D**) correspond to one example rat. (**E**) Explained variance for firing rates, correlations, and BC values, averaged over datasets. Error bars indicate SEM. (**F**) Correlation between firing rates and BC values, averaged over datasets. Error bars indicate SEM.

Despite being independent during active periods, the units in the NHPP data were correlated due to common global fluctuations across epochs and silent and active periods. Indeed, the mean correlation values in each epoch, $\langle C \rangle_e$, from the original and the surrogate data were highly correlated (**Appendix 1—figure 2B,E**), although $\langle C \rangle_e$ was systematically lower in the surrogate data than in the original data. We tested how well this simple model predicted the structure of correlations, by calculating the pairwise correlations, $C_{ij}$, and the BC values in the surrogate data, and comparing them to the values obtained in the original data (**Appendix 1—figure 2C-D**). We found that neither $C_{ij}$ nor BC were predicted by the NHPP: the average fraction of explained variance was equal to $R^2 = 0.148 \pm 0.07$ and $R^2 = 0.184 \pm 0.07$ for $C_{ij}$ and BC, respectively (**Appendix 1—figure 2E**). Moreover, the correlation between firing rates and BC values was substantially reduced in the NHPP, with correlation coefficients equal to $0.59 \pm 0.11$ and $0.19 \pm 0.15$ for the original and the NHPP, respectively ($p < 0.01$, $t$-test). We concluded that correlations and centrality measures were not fully predicted by globally modulated firing rates but were rather suggestive of functional interactions.

