## [Decision Letter]

**Acceptance summary:**

This study shows that auditory cortex dynamics can be decomposed into a small set of critical dimensions and a set of dimensions that rather account for small variations of the dynamics. In addition, remarkably, critical dimensions are not shared between spontaneous and evoked activity, suggesting that the two types of activity come from different processes. This is an exciting result that should motivate important refinements in current models of auditory cortex dynamics.

**Decision letter after peer review:**

Thank you for submitting your article "Cortical state transitions and stimulus response evolve along stiff and sloppy parameter dimensions, respectively" for consideration by *eLife*. Your article has been reviewed by three peer reviewers, including Brice Bathellier as the Reviewing Editor and Reviewer #1, and the evaluation has been overseen by Michael Frank as the Senior Editor. The following individuals involved in review of your submission have agreed to reveal their identity: Matthias H. Hennig (Reviewer #2); Mark D Humphries (Reviewer #3).

The reviewers have discussed the reviews with one another and the Reviewing Editor has drafted this decision to help you prepare a revised submission.

Summary:

This paper uses model-based analysis of extracellular recordings from the rat auditory cortex to provide evidence for the hypothesis that the stability of the population in cortical networks depends on a small number of neurons with highly sensitive parameters, while parameters associated with the majority of neurons are insensitive to changes. Moreover, the authors show that transitions between desynchronised and synchronised states are associated with changes in the sensitive parameters, while during sensory stimulation this is state-dependent. Finally, it is shown that neurons with sensitive parameters have hub-like properties in the network.

Overall these are very interesting and in fact surprising results. The paper is well written and presented, but a few complementary analyses/clarifications are needed to make the important but complex message of this study more accessible and convincing.

Essential revisions:

1) The central conclusions in this paper are based on analysis of the Fisher information matrix (FIM) for pairwise maximum entropy models fit to binarised multi-neuron patterns. One possible caveat here is that the models were fit on relatively small data sets, which could affect the analysis downstream as the insensitive directions in the FIM could just reflect poor parameter estimates. The around 5000 patterns used here to fit the model are very few, and estimates of correlations used to constrain the model may be poor, given cortical activity is quite sparse. Possible biases should be assessed more thoroughly. In particular it is important to check for the impact of estimation errors given the short epochs. One key measure to check is the sensitivity measure, as this defines both core results (spontaneous = stiff; evoked = sloppy). The authors should provide an estimate of the variation in "sensitivity" as a function of model estimation error.

Most importantly:

– Figure 5E-G, check for how correlations between sensitivity and "state" change with errors in model estimation.

– Figure 6D-E, check for how correlations between sensitivity and the modulation index change with errors in model estimation.

2) It is hard to follow what is 'stiff' and what is 'sloppy', as in most of the cases the frontier between the two categories is obviously not sharp. It is clear that there are two ends of the parameter distribution (one end is more stiff and the other more sloppy), but in between, there are many intermediates. The authors must be more explicit about this and make more precise wherever possible what they mean when they define these two categories.

– In Figure 4B they should display the boundary between stiff and sloppy with a shading or a vertical line. Also, there is a kink in the eigenvalue curve. Is it related to model/population size? Are the eigenvalues beyond this kink relevant?

– It should be explained why stiffness of individual neurons is chosen to be their contribution to the first eigenvalue only. Are further eigenvalues not relevant?

– Together it should be clarified in a dedicated paragraph that stiff and sloppy parameters/neurons are two parts of a wide distribution.

---

## [Author Response]

Essential revisions:1) The central conclusions in this paper are based on analysis of the Fisher information matrix (FIM) for pairwise maximum entropy models fit to binarised multi-neuron patterns. One possible caveat here is that the models were fit on relatively small data sets, which could affect the analysis downstream as the insensitive directions in the FIM could just reflect poor parameter estimates. The around 5000 patterns used here to fit the model are very few, and estimates of correlations used to constrain the model may be poor, given cortical activity is quite sparse. Possible biases should be assessed more thoroughly. In particular it is important to check for the impact of estimation errors given the short epochs. One key measure to check is the sensitivity measure, as this defines both core results (spontaneous = stiff; evoked = sloppy). The authors should provide an estimate of the variation in "sensitivity" as a function of model estimation error.

As suggested by the reviewers, in the revised manuscript, we now evaluated the correlation between sensitivity and the error on the model estimation of the firing rates and covariances. This analysis is mentioned in the main article in subsection “Sensory-evoked activity evolves along sloppy dimensions” and described in detail in the Appendix 1 and in Appendix—figure 1.

The estimated firing rates and covariances can be calculated from the model parameters by means of the model's Boltzmann distribution. The firing rates and covariances estimated from the data and those estimated by the MEM highly correlated (average correlation: 0.999 ± 0.001 for rates; 0.956 ± 0.018, for covariances), indicating acceptable fits of the models (Appendix—figure 1A-C). This goes in line with the reported high Kullback-Leibler ratios (0.95 ± 0.03 on average) that quantify model performances (Appendix—figure 1D).

We next defined the estimation errors as the averaged normalized squared difference between the values estimated from the data and those estimated by the model. We found a correlation between the estimation and sensitivity of 0.192 ± 0.055 (Appendix—figure 1E). We next asked whether the correlations between sensitivity and cortical state observed in Figure 5F could be fully predicted by correlations between errors and cortical state. We found that correlations between sensitivity and were higher than correlations between estimation errors and (0.252 ± 0.056 vs. 0.168 ± 0.017, Appendix—figure 1F). We used Meng's z-test for dependent correlations to compare the correlations between sensitivity and those between estimation errors and, given the correlation between and errors, and found significant differences in 4/6 datasets (p < 0.05). This was the case for datasets with the highest model performances (datasets 1–4). Finally, we tested whether estimation errors for firing rates correlated with the modulation index (MI) of the neurons and compared the results with those obtained in Figure 6D. We found that correlations between estimation errors and MI were substantially weaker than correlations between estimation errors and MI (-0.506 ± 0.043 vs. -0.086 ± 0.017 on average; p < 0.05 for all datasets, Meng's z-test; Appendix—figure 1G).

We concluded that, although estimation errors correlated with sensitivity, they did not fully explain neither the correlation between sensitivity and cortical state (for the datasets with highest model performance) nor the correlation between sensitivity and MI.

Most importantly:– Figure 5E-G, check for how correlations between sensitivity and "state" change with errors in model estimation.

As mentioned above, estimation errors did not fully explain the correlation between sensitivity and cortical state.

– Figure 6D-E, check for how correlations between sensitivity and the modulation index change with errors in model estimation.

As mentioned above, estimation errors did not fully explain neither the correlation between sensitivity and MI.

2) It is hard to follow what is 'stiff' and what is 'sloppy', as in most of the cases the frontier between the two categories is obviously not sharp. It is clear that there are two ends of the parameter distribution (one end is more stiff and the other more sloppy), but in between, there are many intermediates. The authors must be more explicit about this and make more precise wherever possible what they mean when they define these two categories.– In Figure 4B they should display the boundary between stiff and sloppy with a shading or a vertical line.

Stiff and sloppy dimensions are those associated with low and high rank eingenvectors of the FIM, respectively. In sloppy systems there is no clear cut between stiff and sloppy dimensions, so it is not possible to draw a line separating stiff and sloppy dimensions, thus we added a shading in Figure 4B. We concentrated into the first eigenvector to estimate the sensitivity of parameters, but, as shown below, we now also provide a more general definition of sensitivity that considers the weighted contribution to all eigenvectors (see below).

Also, there is a kink in the eigenvalue curve. Is it related to model/population size? Are the eigenvalues beyond this kink relevant?

In our study, the stiff dimension was defined as being the first eigenvector of the FIM (see next point to a discussion about an alternative definition). The kink in the eigenvalue curve observed in Figure 4B was observed in two datasets, for rat 1 (which is presented in the figure) and rat 2. For the other datasets, this kink was not clearly observed as shown Author response image 1. Thus, this observation is not systematic in our sample. The relation between the kink observed in rats 1 and 2 and the size of the population is not straightforward. One could think that the first eigenvectors relate to the bias parameters of the neurons, but, as shown in Figure 5D (bottom), biases and couplings contributed equally to the first eigenvector. We believe that a discussion about the kink observed in those datasets and its relation to model size is out of the scope, since our results are not based on this part of the spectrum but mainly focus on the first eigenvector. Concerning the relevance of eigenvectors beyond the first one, see next point.

– It should be explained why stiffness of individual neurons is chosen to be their contribution to the first eigenvalue only. Are further eigenvalues not relevant?

Following the work of Panas et al., 2015, we have measured the sensitivity of a given model parameter by its contribution to the first eigenvector of the FIM averaged across time. However, it could happen that the parameters that contribute less to the first eigenvector do contribute to the other stiff dimensions (those with lower rank, for example 2). To take into account this, in the new manuscript, we added an alternative definition of sensitivity that considers the weighted contribution to all eigenvectors of the FIM. Specifically, for each 100-s epoch and each neuronal ensemble, we calculated the eigenvectors of the FIM (*v_t,_*_1_,…,*v_t,_*_55_) and their associated eigenvalues (*a_t_*_,1_,…,*a_t_*_,55_). We defined the weighted sensitivity of the parameter as the temporal average of its weighted contribution to all eigenvectors of the FIM, with weights equal to the associated eigenvalues:

sne,iw=1NE∑t=1NE∑k=155at,k|Vt,ki|at,1+at,2+at,55

Just as for the original definition, one can construct a weighted sensitivity at the population level.

Using this more general definition, all the results were preserved. This version of the sensitivity is described in Materials and methods (subsection “Sensitivity measures”) and the results are described in Figure 5—figure supplement 1, Figure 6—figure supplement 1, Figure 7—figure supplement 1, and in the main article in subsections “Cortical state transitions evolve along stiff dimensions”, “Sensory-evoked activity evolves along sloppy dimensions” and “Stiff parameters were associated to central neurons within the neuronal network”.

– Together it should be clarified in a dedicated paragraph that stiff and sloppy parameters/neurons are two parts of a wide distribution.

We explicitly mention this point in the discussion of the revised manuscript. See paragraph one and five in the Discussion.